# Pianno: a probabilistic framework automating semantic annotation for spatial transcriptomics

Yuqiu Zhou[1], Wei He[1], Weizhen Hou[1] & Ying Zhu ●[1] ✉

Spatial transcriptomics has revolutionized the study of gene expression within tissues, while preserving spatial context. However, annotating spatial spots' biological identity remains a challenge. To tackle this, we introduce Pianno, a Bayesian framework automating structural semantics annotation based on marker genes. Comprehensive evaluations underscore Pianno's remarkable prowess in precisely annotating a wide array of spatial semantics, ranging from diverse anatomical structures to intricate tumor microenvironments, as well as in estimating cell type distributions, across data generated from various spatial transcriptomics platforms. Furthermore, Pianno, in conjunction with clustering approaches, uncovers a region- and species-specific excitatory neuron subtype in the deep layer 3 of the human neocortex, shedding light on cellular evolution in the human neocortex. Overall, Pianno equips researchers with a robust and efficient tool for annotating diverse biological structures, offering new perspectives on spatial transcriptomics data.

Recent advancements in spatial transcriptomics techniques such as 10× Visium[1], Slide-seq[2], and Stereo-seq[3], have revolutionized the study of gene expression patterns within tissues while preserving their spatial information. However, merely obtaining gene expression profiles at specific physical coordinates within a tissue is insufficient to fully understand the complexity of biological systems. To gain deeper insights, it is imperative to discern the biological identity of each spatial spot within the tissue, a process referred to as pattern annotation[4]. These patterns can represent brain regions, tumor or normal tissue, and cell types. This concept mirrors the idea of "semantic segmentation" in computer vision, where pixels are categorized into patterns to elucidate visual content[5,6]. In light of this analogy, we introduce the concept of spatial transcriptomics semantic annotation, which assigns spatial spots within tissue to patterns of predefined structures or cell types. By implementing this concept, our primary objective is to enhance the interpretation of intricate biological systems by incorporating information from multiple dimensions.

For biological interpretation of spatial transcriptomics, many machine learning-based approaches have been developed to identify clusters of spatial units (spots) and interpret their biological

identities using marker genes[7–11]. However, despite incorporating spatial information, these approaches typically yield clusters that primarily consist of groups of transcriptomically similar spots, thereby lacking the ability to establish a clear connection between these clusters and the known structures. Taking the human neocortex as an example, which is known to consist of gray matter comprising six layers and white matter, representing the anatomical level patterns[12,13]. On the other hand, it comprises different types of neurons and glia, representing the cell-type level patterns[14,15]. Data-driven clustering methods may not always be able to identify all neocortical layers or cell types. Instead, they may identify clusters representing different levels of patterns, i.e., some clusters based on layers and some others based on cell types. In some cases, these methods may even group spatial spots together due to the combined effects of layers and cell types. While this feature may assist in the identification of novel structures, additional procedures are necessary to supplement their biological interpretation. Manual annotation is often employed as a supplementary approach to label known anatomical structures and aid in the interpretation of unsupervised clusters[16]. However, manual annotation heavily relies on the

[1]State Key Laboratory of Medical Neurobiology, MOE Frontiers Center for Brain Science, Institutes of Brain Science and Department of Neurosurgery, Huashan Hospital, Fudan University, Shanghai, China. ✉e-mail: ying_zhu@fudan.edu.cn

researchers' expertise, introducing subjectivity and posing challenges when attempting to scale up in large-scale analyses.

At the cell-type level, commonly used tools to explore the spatial distribution of cell types relies heavily on deconvolution approaches that establish a mapping between single-cell and spatial transcriptomics data[17–21]. However, these tools are constrained by the requirement of single-cell RNA-seq data from the same tissue and the potential interference from batch effects. Recently, marker-gene-driven approaches started to emerge for cell segmentation of multiplexed in situ imaging data[22], and for cell-type deconvolution in spatial transcriptomics[23]. However, there is still a lack of marker-gene-driven spatial semantic annotation tools.

To address the limitations in existing approaches, we developed Pianno (Pattern image annotation), a Bayesian framework that automatically annotates biological identity of spots in spatial transcriptomics using pre-defined marker lists. Pianno has the unique capability to automatically label patterns, including both anatomical structures and cell types, with just a few marker genes. This framework is applicable to data generated by various spatial transcriptomics techniques. In our evaluation, Pianno demonstrated superior performance when compared to state-of-the-art spatial clustering methods, accurately identifying patterns that closely resemble manual labeling. Additionally, Pianno outperformed over deconvolution methods in reconstructing the spatial distribution of cell types. By implementing Pianno, we uncovered brain region- and species-specific spatial expression patterns of neurofilament genes in layers 3 and 5 of the neocortex. Further analysis of these gene expression patterns revealed intriguing insights into regional specification and evolutionary changes of these neurons.

## Results

### Workflow of Pianno

Pianno employs a probabilistic framework to perform semantic annotation on spatial transcriptomics based on a set of marker genes (Fig. 1). The input to Pianno includes spatial transcriptomic data, comprising spatial coordinates, raw gene counts, and an initial marker gene list with as few as one known marker provided for each pattern. The annotation process consists of two sequential steps: the initial segmentation step and the refinement step. In the initial step, each gene's spatial expression is transformed into a grayscale image. Then, for each target pattern, a pattern image is created by aggregating the grayscale images of the marker genes associated with that pattern. The initial marker list is then updated by identifying additional candidate marker genes for each pattern, considering their distinct expression patterns across the initially annotated structures. This refined marker list is integrated into the subsequent refinement step. Within this refinement stage, a Bayesian classifier is built to estimate the posterior probability of each spatial spot belonging to different patterns. The annotation is then updated based on the posterior probabilities. Pianno offers two methods for updating the annotation. For continuous patterns in semantic annotation, it is recommended to take the probability distribution as a pattern image and return it to the pattern detector for updated annotation. For dispersed or sharp-shaped patterns, directly updating the label based on the probability value is recommended, as it preserves detailed information. Overall, Pianno not only streamlines the annotation process but also employs a heuristic approach to identify additional marker genes using the initial single marker gene, thereby minimizing the requirement for the number of known markers as input. Further details can be found in the Methods.

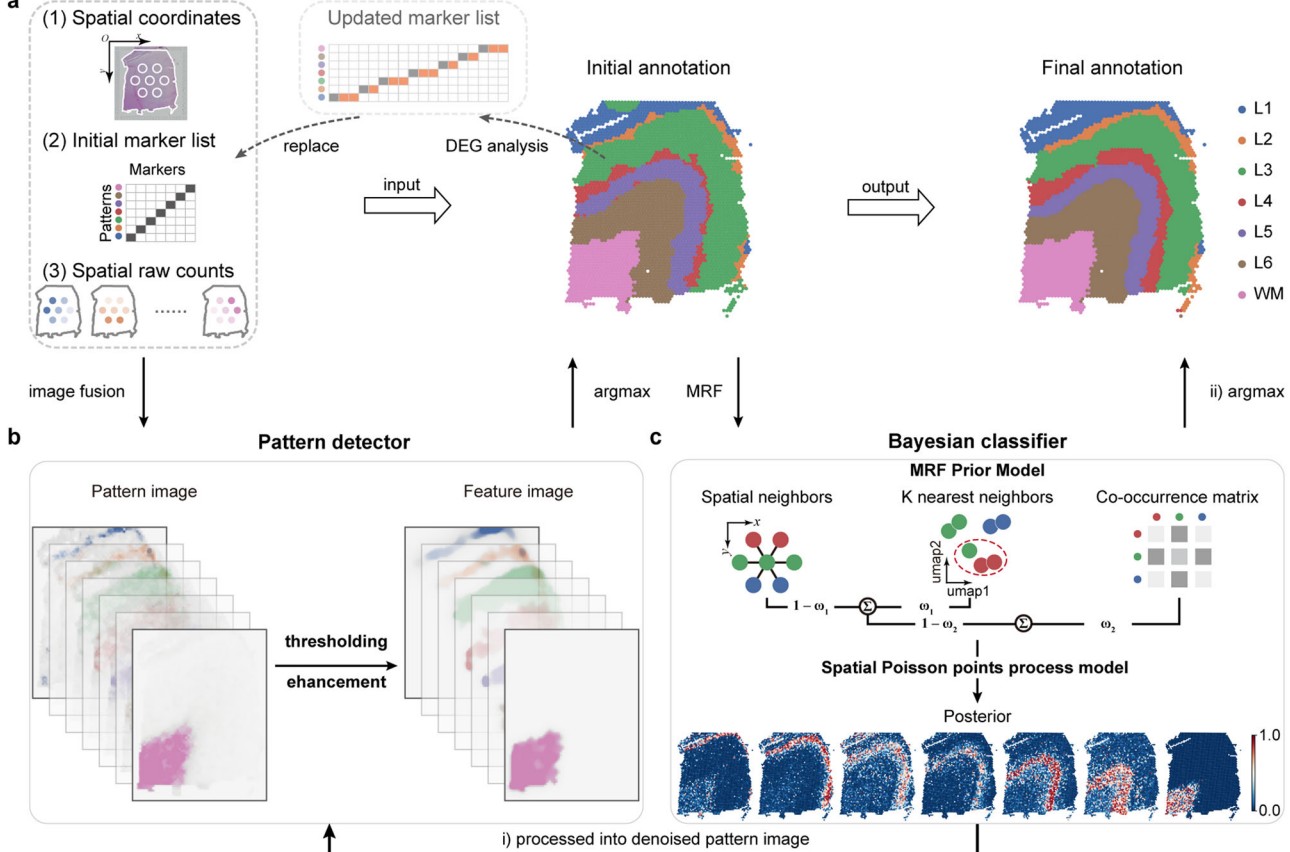

**Fig. 1 | Overview of Pianno. a** Pianno's inputs include spatial coordinates, an initial marker gene list with as few as one known marker labeling each pattern, and raw gene counts. DEG differentially expressed gene. **b** Workflow of pattern detector. **c** Workflow of Bayesian classifier. MRF Markov random field.

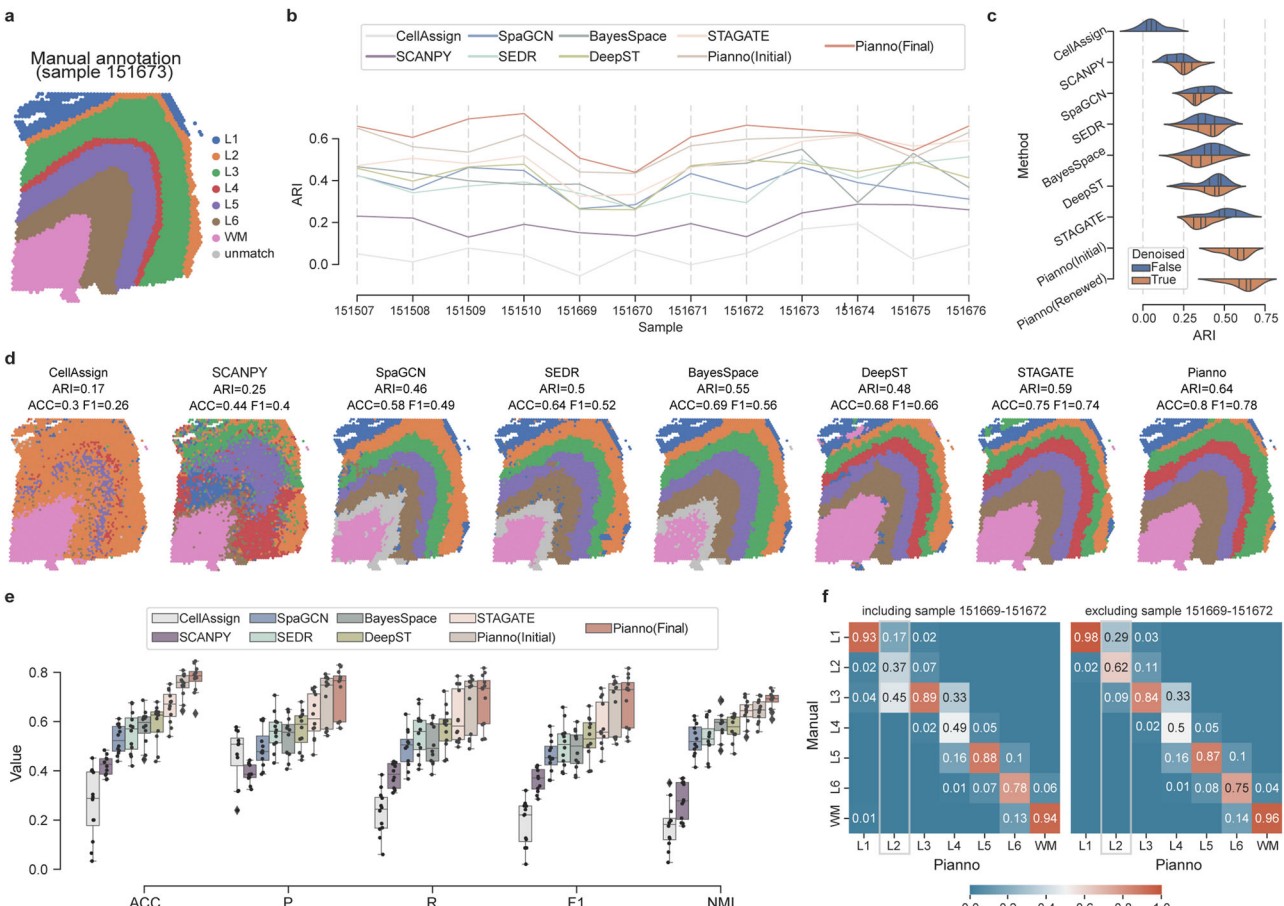

**Fig. 2 | Evaluation of Pianno's performance in cortical architecture reconstruction. a** Manual annotation of anatomical structures, including cortical layers (L1-L6) and white matter (WM), within a representative dlPFC section (sample 151673). **b** Adjusted rand index (ARI) assessing the concordance between labels predicted by different methods and the manual annotation. The black lines inside the violin indicate the quartiles. **c** Comparison of dlPFC samples (n=12) with and without SAVER preprocessing. **d** Cortical architecture segmentation by CellAssign[24], SCANPY[25], SpaGCN[8], SEDR[9], BayesSpace[7], DeepST[10], STAGATE[11], and Pianno. The clusters identified by spatial clustering approaches were mapped to the manual annotation using Kuhn-Munkres algorithm[59]. **e** Boxplot summarizing

annotation metrics across all 12 samples. The box bounds the interquartile range (IQR) divided by the median, with the whiskers extending to a maximum of 1.5 × IQR from the box, and values beyond the whiskers are considered outliers, marked with diamonds. ACC: accuracy; P: macro-averaging precision; R: macro-averaging recall; F1: macro-averaging F1-score; NMI: normalized mutual information. **f** Confusion matrix normalized by column depicting the comparison between Pianno and manual annotations in all 12 samples (including sample 151669–151672, left) and 8 samples (excluding sample 151669–151672, right). Diagonal values in the matrix represent the precision of each layer. Source data are provided as a Source Data file.

## Superiority of Pianno over clustering-based tools in anatomical structure annotation

The performance of Pianno was first evaluated using 12 samples from the human dorsolateral prefrontal cortex (dlPFC)[16], and compared to another marker-based but non-spatial-aware annotation method CellAssign[24]. The widely-used unsupervised clustering approach, Leiden algorithm[25], and five widely-used spatial clustering methods (SpaGCN[8], SEDR[9], BayesSpace[7], DeepST[10], and STAGATE[11], Supplementary Fig. S1) were also considered in the evaluation process. In order to match the known layers in the human neocortex, the number of clusters was set to seven. Pianno demonstrated the highest agreement with manual annotation conducted by experienced researchers based on morphological features and markers[16] (Fig. 2a), outperforming all other tested methods in 11 out of 12 samples, as indicated by the Adjusted Rand Index (ARI)[26] (Fig. 2b). Our analysis also highlighted the pivotal role of the Bayesian classifier in refining the initial annotations generated within the Pianno pipeline by the pattern detector (Fig. 2b). Notably, even the initial annotations remained superior to other methods according to ARI (Fig. 2b, e). Furthermore, Pianno's superior performance was reaffirmed through comprehensive evaluation using additional classification metrics, including

accuracy (ACC), macro-averaging precision (P), macro-averaging recall (R), macro-averaging F1-score (F1), and normalized mutual information (NMI) (Fig. 2e, Supplementary Fig. S2, Supplementary Data 1).

Apparently, non-spatial-aware marker-based or clustering methods have difficulties in recognizing spatial domains (Fig. 2d). While Pianno consistently exhibited high performance across the diverse samples, clustering-based tools occasionally encountered challenges in identifying all known structures accurately. For instance, BayesSpace and SEDR failed to identify clusters that resembled layer-like structures in certain samples (Fig. 2d and Supplementary Fig. S2). Some other tools identified clusters that did not correspond precisely to known neocortical layers, causing issues such as missing annotations (Fig. 2d, e.g., SpaGCN, BayesSpace, and DeepST) or one-to-many mappings (Fig. 2d and Supplementary Fig. S2), ultimately resulting in matching ambiguities.

As Pianno adopts SAVER for denoising spatial transcriptome data in the preprocessing step, we compared Pianno's performance with other approaches (except CellAssign which only accepts raw counts as input) (Fig. 2c), both with and without SAVER preprocessing applied to the data. We observed that except for SCANPY, other methods using denoised inputs do not improve the clustering accuracy, and are even

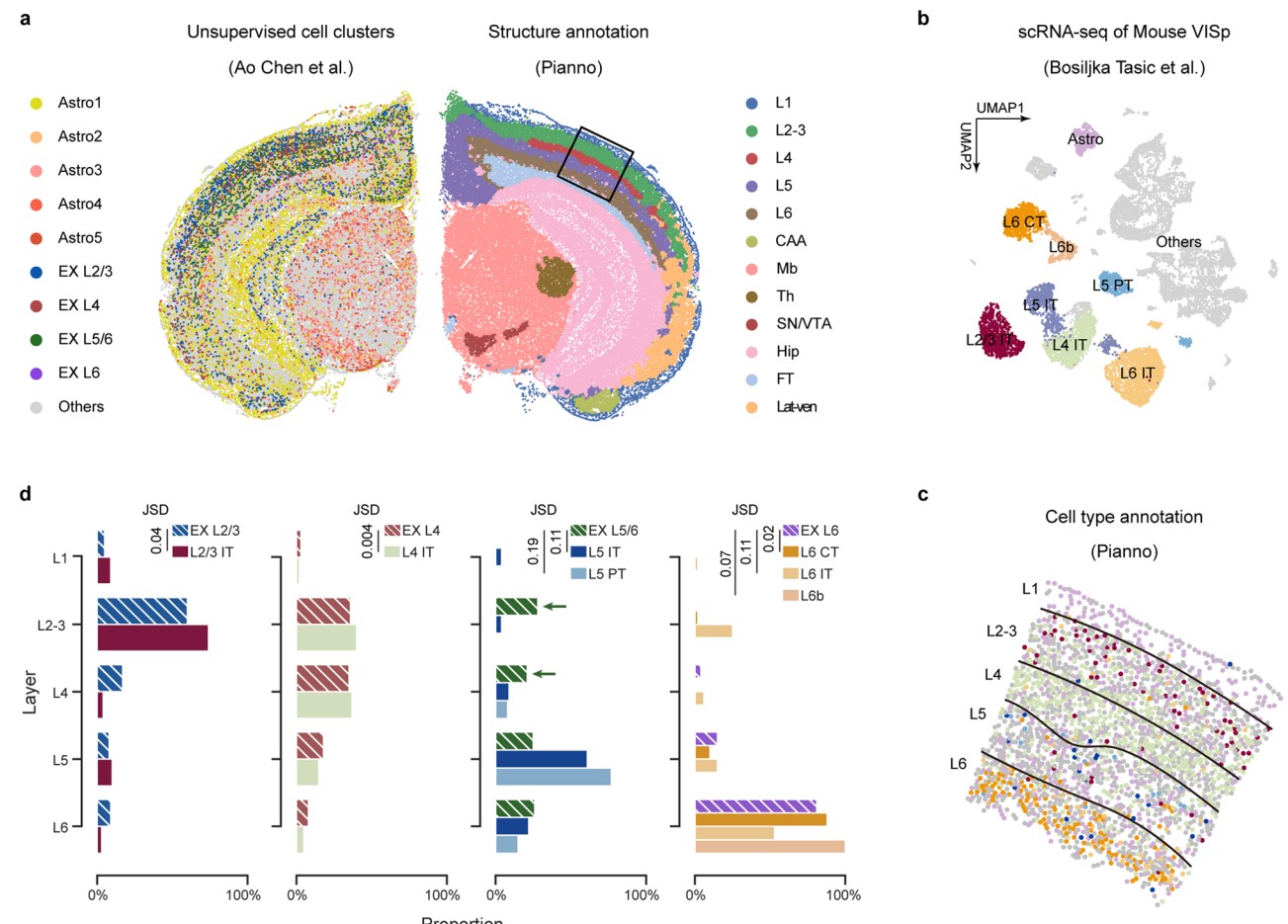

**Fig. 3 | Benchmarking Pianno's performance in cell type annotation within the mouse cortex. a** Visualization of the Stereo-seq dataset from an adult mouse coronal hemibrain[3]. Left: cell bin followed by unsupervised clustering revealed distinct cell type clusters, including astrocytes (Astro); excitatory neurons (EX) and others. Right: Pianno provides a detailed structural annotation of cortical layer (L1-L6), cortical amygdala area (CAA), midbrain (Mb), thalamus (Th), substantia nigra/ventral tegmental area (SN/VTA), hippocampal region (Hip), fiber tract (FT), and lateral-ventral cortex (Lat-ven). The black box highlights the VISp selected as the ROI. **b** UMAP visualization of single-cell RNA-seq data from the mouse VISp colored by cell type[30]. Labeled cell types include astrocytes (Astro), as well as intratelencephalic (IT), pyramidal tract (PT) and corticothalamic (CT) neurons. **c** Spatial distribution of mouse VISp cell types annotated by Pianno using markers identified from the single-cell RNA-seq data in panel **b**. **d** Bar plot summarizing the frequency distribution of excitatory neurons on each layer annotated by unsupervised clustering (striped bar) and Pianno (solid bar), respectively. The Jensen-Shannon divergence (JSD) quantifies the dissimilarity between the two distributions, with values ranging from 0 to 1. Closer to 0 indicates greater similarity. Source data are provided as a Source Data file.

worse than using raw counts (Fig. 2c, e.g., STAGATE, SpaGCN). Pianno consistently demonstrated superior performance compared to the other tools overall. In terms of Pianno's performance at the individual layer level, we observed that the precision of layer 2 (L2) and layer 4 (L4) was relatively lower compared to other layers (Fig. 2f). Upon closer inspection, we found that Pianno labeled L2 in four samples (151669-151672), in which L2 could not be distinguished by manual annotation. The lower accuracy in labeling L4 can be attributed to its continuity with L3 (Supplementary Fig. S3). These observations will be extensively explored in the final section, wherein Pianno is coupled with an unsupervised approach and multimodal data to delve deeper into the underlying biological intricacies.

Together, the benchmarking results demonstrated Pianno's superior capability in accurately identifying and establishing correspondence with anatomical structures, surpassing the performance of the present spatial and non-spatial-aware methods.

**Accurate estimation of cell-type spatial distribution by Pianno**
The ability of Pianno to estimate cell-type spatial distribution was then assessed using a Stereo-seq dataset of an adult mouse hemibrain coronal section[3], and the results were compared with the cell type distribution inferred through distinct strategies: cell segmentation followed by unsupervised clustering, and three widely-used tools for spatial deconvolution based on spatial and single-cell transcriptomics integration[18] (Fig. 3a). Given the well-established knowledge of the cell type composition in the mouse primary visual cortex (VISp) from previous studies and the cumulated single-cell RNA-seq data from this region[27–30], we first employed Pianno to annotate the anatomical structure of the hemibrain and subsequently focused on the VISp region as our region of interest (ROI) (Fig. 3a). To define cell-type-specific markers for Pianno, we utilized a well-annotated mouse VISp single-cell dataset[30] (Fig. 3b). This dataset was also adopted as a reference for the deconvolution methods. Our specific investigation centered around the spatial distribution of excitatory neurons, which are known to display laminar distribution[31,32] (Fig. 3a, b).

We found that Pianno's prediction of the distribution of excitatory neuron subtypes displayed similar pattern as Tangram[17] and RCTD[33] (Supplementary Fig. S3), exhibited a high level of consistency with their known locations across the various layers (Fig. 3c). In contrast, the spatial distribution obtained from unsupervised clustering of cell bins in the original paper presented clusters containing neurons from

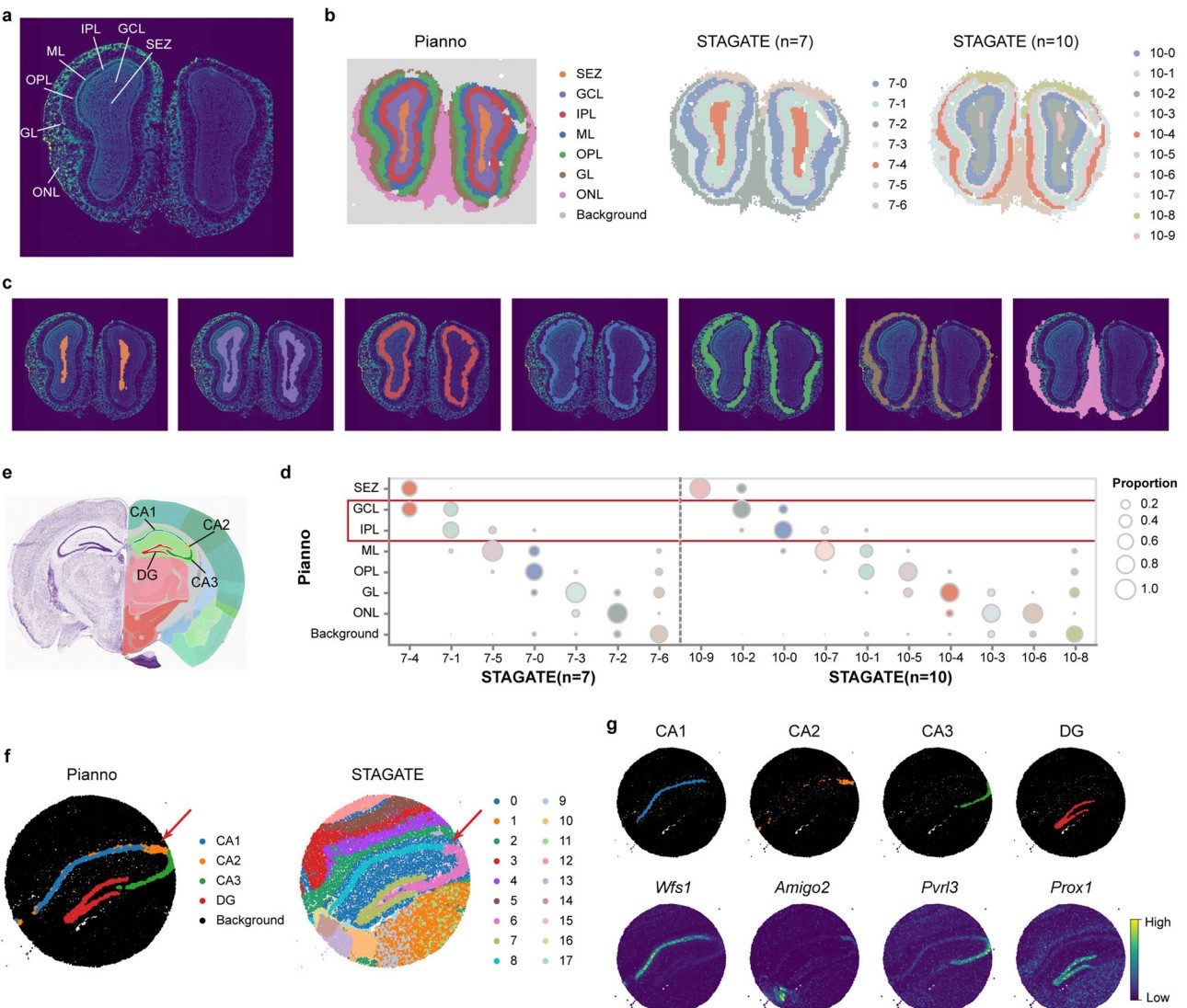

**Fig. 4 | The performance of Pianno in annotating various-shaped structures across different platforms. a** 4',6-diamidino-2-phenylindole (DAPI) staining of a mouse olfactory bulb (MOB) analyzed by Stereo-seq[3]. **b** MOB structure inferred by Pianno and STAGATE respectively. The 'n' denotes the number of unsupervised clusters obtained by STAGATE. Annotated structures include subependymal layer/subventricular zone (SEZ), granule cell layer (GCL), internal plexiform layer (IPL), mitral cell layer (ML), outer plexiform layer (OPL), glomerular layer (GL), and olfactory nerve layer (ONL). **c** Alignment of each MOB structure annotated by Pianno with DAPI-stained images. **d** Proportion of spatial domains obtained by STAGATE in each structure for *n* = 7 and *n* = 10, respectively. **e** Anatomical structure of the hippocampus obtained from the Allen Mouse Brain Atlas. **f** Subregions in hippocampus annotated by Pianno and STAGATE respectively. **g** Visualization of the hippocampal subregions annotated by Pianno and the expression patterns of corresponding markers. Source data are provided as a Source Data file.

both L5/6 and L2-3. These clusters included neurons assigned as EX L5/6 but displayed a significant enrichment also in upper layers (Fig. 3d). Ectopic mapping of L5/6 neurons to L2-3 was also observed in the results obtained from Cell2location[18] (Supplementary Fig. S3), the tool showing top-performance in a previous benchmarking study of spatial deconvolution methods[34]. These results suggest that even the most advanced deconvolution tool available at the time struggled with precisely assigning these neurons to their correct layers. The mispositioning of L5/6 neurons to L2-3 can likely be attributed to the fact that L2-3 and L5 pyramidal neurons share some similar electrophysiological properties, morphological features, and connectivity patterns, leading to an increased transcriptomic similarity between them[35,36].

Overall, our results demonstrated the robustness and accuracy of Pianno in estimating intricate cell-type distributions in spatial datasets, particularly in situations where unsupervised approaches encounter challenges.

## Effective annotation of complex structures on diverse spatial transcriptomics platforms

In addition to the neocortex, we further assessed the performance of Pianno in annotating variously shaped structures in spatial transcriptomics data from different platforms. Pianno's performance was compared with that of STAGATE[11], which exhibited the highest performance in the aforementioned benchmarking. Through these evaluations, we demonstrated the effectiveness and efficiency of Pianno.

Firstly, we applied Pianno to annotate anatomical structures in a Stereo-seq dataset of the mouse olfactory bulb (MOB)[3] (Fig. 4a). The dataset contains 10,747 spatial spots, including tissue-covered areas and the background area. Pianno was able to complete both background removal as well as structure annotation at the same time within a couple of minutes (Fig. 4b). The structures identified by Pianno displayed good correspondence with known anatomical structures based on cytoarchitecture and marker genes (Fig. 4c, Supplementary Fig. S4). In contrast, STAGATE failed to identify clusters corresponding to all

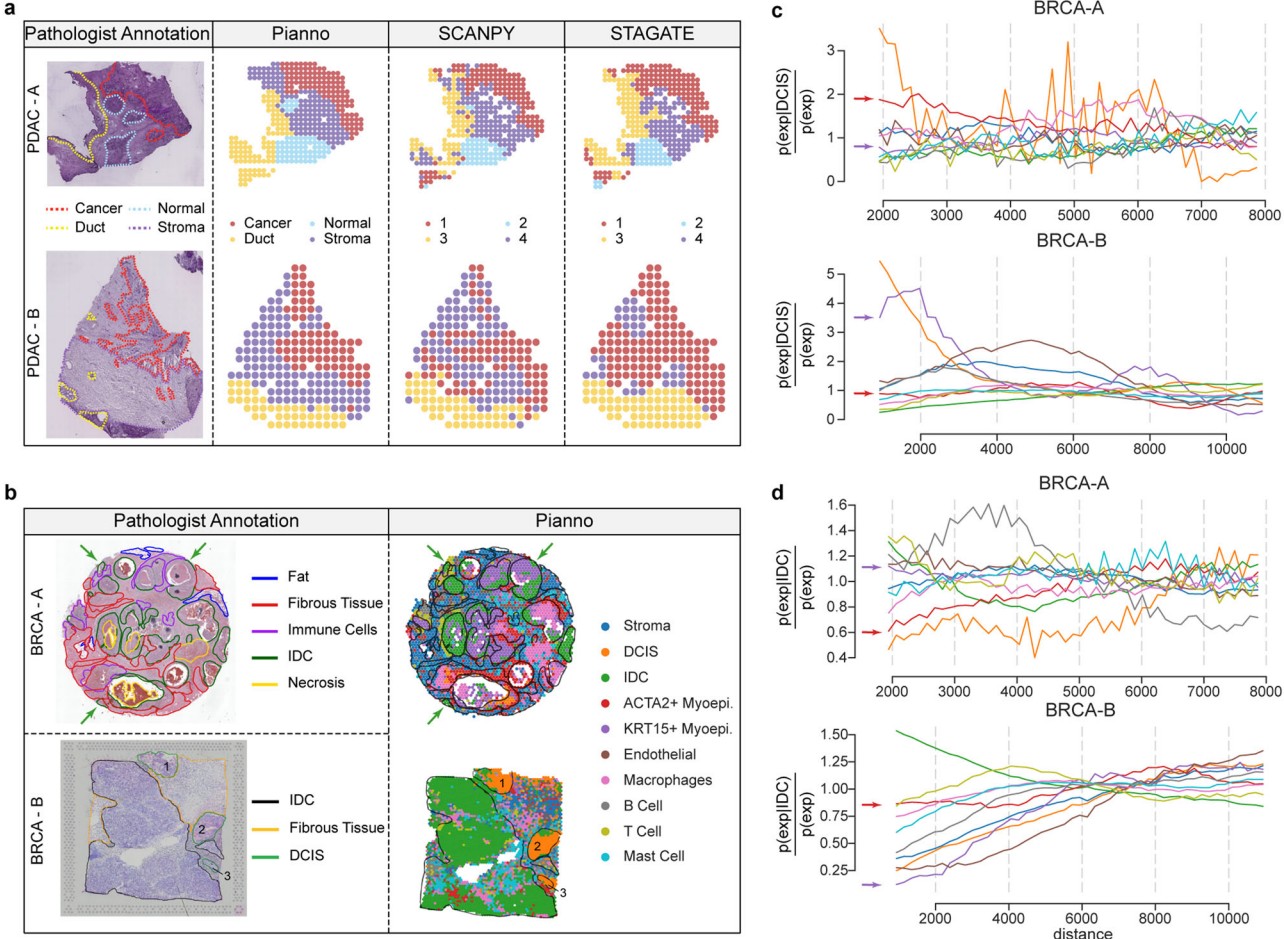

**Fig. 5 | Annotation of tumor microenvironments. a** Microarray-based spatial transcriptomics (ST) of two human pancreatic ductal adenocarcinoma samples[39], PDAC-A (top) and PDAC-B (bottom). From left to right, the panels display pathologist annotations on hematoxylin and eosin (H& E) images, Pianno's annotations, unsupervised clustering by SCANPY, and clusters identified by STAGATE incorporating spatial information, respectively. **b** Annotations of two breast cancer samples, BRCA-A (top) and BRCA-B (bottom). From left to right, panels display pathologist annotations on H& E images and Pianno annotations, respectively. DCIS, ductal carcinoma in situ; IDC, invasive carcinoma; Myoepi., myoepithelial. **c** Co-occurrence score between DCIS and the rest of the cell types in BRCA-A (top) and BRAC-B (bottom). **d** Co-occurrence score between IDC and the rest of the cell types in BRCA-A (top) and BRAC-B (bottom). Source data are provided as a Source Data file.

anatomical structures when setting the cluster number equal to the number of structures (seven). While it managed to identify most structures, it was not able to distinguish the granule cell layer (GCL) from the internal plexiform layer (IPL). Moreover, it had an additional cluster likely representing blood contamination (Cluster 7-6). Increasing the cluster number to 10 improved the detection of known anatomical structures, with GCL identified from IPL. However, it also included some clusters that were challenging to assign a specific structural identity, potentially representing blood contamination (Cluster 10-8) or background noise (Cluster 10-6) (Fig. 4d).

Additionally, we assessed the implementation of Pianno to annotate subregions, namely field CA1, CA2, CA3, and DG (dentate gyrus), within the mouse hippocampus using data from a higher resolution spatial transcriptomics platform Slide-seq V2[37] (Fig. 4e). Unsupervised approaches, such as STAGATE, failed to locate the filed CA2 even when the number of clusters was increased to 18 (Fig. 4f). In contrast, Pianno successfully identified all subregions, corresponding to anatomical structures defined in reference brain atlases based on cytoarchitecture and marker expressions (Fig. 4g).

Overall, these results highlight the advancement of Pianno in annotating anatomical structures of interest. While unsupervised clustering approaches identify clusters based on transcriptome similarity, potentially including some clusters that are challenging to

interpret with prior knowledge, Pianno provides a valuable tool for precisely locating desired anatomical structures.

**Accurate semantic annotation of complex microenvironments**

While the previous sections focused on tissues with continuous structures, the tumor microenvironment presents a highly heterogeneous landscape, comprising a mixture of immune cells, stromal, blood vessels, and extracellular matrix, spread throughout the tissue[38]. To assess Pianno's performance in annotating tissues with such complex and dispersed structures, we applied the tool to analyze the microenvironments of two human pancreatic ductal adenocarcinomas (PDAC) samples and two breast cancer (BRCA) samples[39]. Since the identification of stroma markers is challenging given the complex composition of stromal tissue, we refrained from specifying stroma markers in our analysis. Instead, we designated spots assigned to the "undefined" category as stromal regions. Pianno accurately inferred patterns within the PDAC samples that closely aligned with manual annotations provided by pathologists based on histology and marker expression (Fig. 5a). Pianno's alignment with the manual labeling surpassed the results obtained from unsupervised clustering methods, including SCANPY[25] and STAGATE.

Invasive ductal carcinoma (IDC) represents the most common type of invasive breast cancer, while ductal carcinoma in situ (DCIS) is

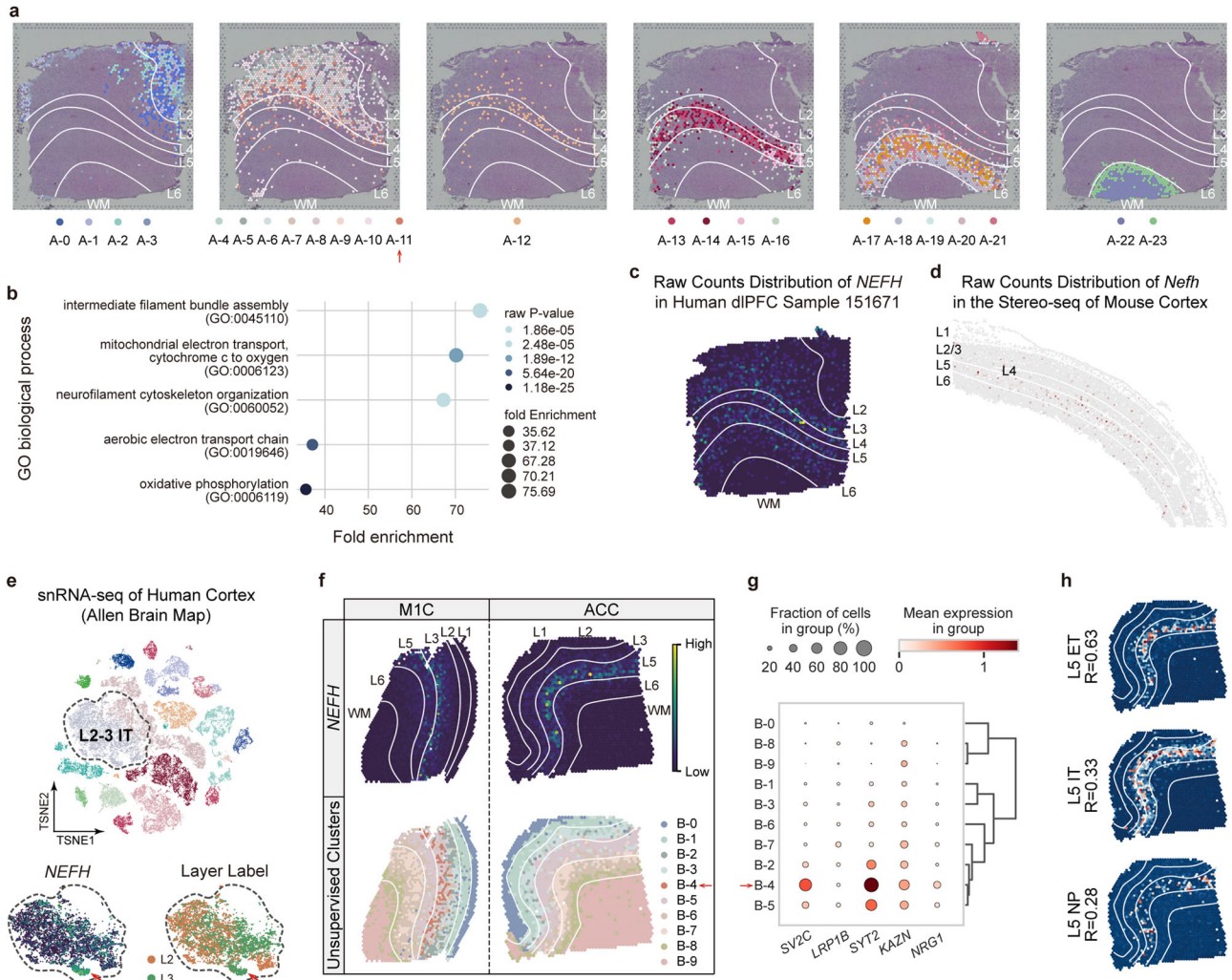

**Fig. 6 | Pianno's semantic annotation uncovering novel regional and species-specific cellular organization. a** Spatial distribution of unsupervised clusters by SCANPY within each structure on the H&E stained image of dlPFC sample 151671. The red arrow pinpoints cluster A-11 in deep L3. **b** Top 5 gene ontology (GO) biological process terms enriched in cluster A-11. The raw P-values are estimated by DAVID website using one-sided Fisher's exact test without adjustment. **c** Visualization of the raw counts of *NEFH* in dlPFC sample 151671. **d** Visualization of the raw counts of *Nefh* in the Stereo-seq dataset of the mouse cortex[3] (as shown in Fig. 3a). **e** Visualization of single-nucleus RNA-seq data from multiple human cortical areas by tSNE (top)[32]. The expression of *NEFH* across L2-3 IT neurons is

highlighted and displayed (bottom). The subcluster with high *NEFH* expression is indicated by the red arrow. **f** Raw counts of *NEFH* (top) and the spatial distribution of unsupervised clusters identified by SCANPY (bottom) in the human primary motor cortex (M1C), and anterior cingulate cortex (ACC). **g** Dot plot depicting the expression of the top 5 marker genes of the *NEFH*-enriched L3 IT subcluster across M1C spatial clusters. The spatial cluster B-4 is highlighted by the red arrow. **h** Pearson correlation coefficients (R) between the *NEFH* expression pattern and the probability distribution of each L5 excitatory neuron subtypes in ACC. ET extra-telencephalic, NP near-projecting neurons. Source data are provided as a Source Data file.

considered a non-invasive or pre-invasive form of breast cancer and is regarded as a precursor to IDC. Understanding the microenvironment in BRCA samples is crucial in determining cancer progression and, consequently, guiding treatment decisions. In our analysis, Pianno identified small areas of IDCs, scattered DCIS regions and a high level of immune cell infiltration in sample A. Conversely, in sample B, IDC covered a large area, with only small DCIS regions and minimal immune cell infiltrations. Notably, Pianno's automatic labeling closely matched the manual annotations made by pathologists (Fig. 5b). Further analysis of cell-type co-occurrence in both BRCA samples revealed a positive correlation between myoepithelial cells and DCIS, but a negative correlation between myoepithelial cells and IDC (Fig. 5c, d). These findings align with previous research that links decreased myoepithelial cells to tumor invasion, indicating potential implications for BRCA diagnosis[40–42].

Overall, our results showcase Pianno's potential in annotating irregular and complex structures, particularly within heterogeneous

tumor microenvironments. These accurate and informative annotations are expected to provide valuable assistance to pathologists in understanding the intricate nature of tumor biology and may hold promise in guiding personalized treatment strategies.

## Revealing new biological insights through multimodal data integration and tool synergy

The aforementioned results demonstrated the complementary strengths and purposes of Pianno and unsupervised clustering approaches. In this section, we showcase how the integration of these two methods facilitates the discovery of new insights into regional and species divergence in cellular diversification within the neocortex.

In the previous section, we noticed a lower precision in the annotation of L2 and L4 by Pianno. This observation prompted us to conduct further exploration. We, therefore, integrated Pianno's annotation with spatial clusters produced by SCANPY in dlPFC samples (Fig. 6a). Taking dlPFC 151671 as an illustrative case, where manual

annotation failed to distinguish L2, we assigned clusters to specific layers based on their predominant layer locations (Supplementary Fig. S5a). This analysis revealed the presence of clusters distinctly positioned in L2 (Clusters A-0, A-1, A-2, A-3). Importantly, Pianno's allocation of L2 labels aligned with the distinct cytoarchitecture of L2, characterized by its composition of smaller and sparser neurons (Supplementary Fig. S5b). Moreover, the high expression of L2 markers further substantiates the accuracy of Pianno's L2 annotations (Supplementary Fig. S5c, d).

However, when considering L4, we did not identify clusters exclusively linked to L4 in the dlPFC spatial transcriptome. Clusters with a substantial presence in L4 also exhibited considerable presence in L3 (Clusters A-11 and A-12), or L5 and L6 (Clusters A-15 and A-16). This observation may provide a plausible explanation for the reduced precision in L4 annotation by Pianno.

Of particular interest, we noted the presence of a cluster (Cluster A-11) in deep L3 (Fig. 6a) enriched with genes associated with functional terms such as intermediate filament bundles and neurofilaments (Supplementary Data 2, 3 and Fig. 6b). Within these terms, a notable member is *NEFH*, which encodes the neurofilament heavy chain, known for its role in the maintenance of neuronal caliber[43,44]. *NEFH* exhibited significant enrichment at the boundary of deep L3 and L4, and a secondary enrichment in L5 (Fig. 6c, Supplementary Fig. S5e). Intriguingly, this pattern was not conserved in the mouse cortex, where the expression of *Nefh* spread across L2-6, with a primary enrichment in L5 (Fig. 6d). These findings aligned with previous immunohistochemistry observations, showing that *NEFH* labels a group of human (or primate)-specific large pyramidal neurons in deep L3 of the secondary visual cortex, whereas, in the mouse neocortex, *Nefh* is predominantly expressed in L5[12,14].

To further explore whether this cross-species differential expression pattern is regional-specific, we generated and analyzed spatial transcriptome data from the human primary motor cortex (M1C) and the anterior cingulate cortex (ACC). Our analysis uncovered a cluster in M1C (Cluster B-4), analogous to cluster A-11 in dlPFC, characterized by a high expression of neurofilament genes and residing in deep L3 (Supplementary Data 4). In contrast, no analogous cluster was identified in L3 of ACC. Furthermore, while the expression pattern of *NEFH* across layers in M1C closely resembled that of dlPFC, in ACC, *NEFH* exhibited higher expression only in L5, without a corresponding presence in deep L3. This regional pattern was further confirmed through in situ hybridization (Supplementary Fig. S5e).

To delve deeper into the identity of neurons expressing *NEFH*, we examined the expression of *NEFH* across neuronal subtypes in single-nucleus RNA-seq (snRNA-seq) data obtained from multiple neocortical regions[32]. We identified a distinct cluster of L2-3 intratelencephalic (IT) neurons that exhibited high *NEFH* expression. The concurrent high expression of other neurofilament genes that were upregulated in the deep L3 spatial cluster of dlPFC and M1C suggested the existence of a subpopulation of long-projection L3 IT neurons exclusive to specific areas in the human neocortex, but not in mice. In contrast, we did not observe a distinct cluster of L5 neurons showing *NEFH* enrichment. Therefore, we calculated the correlation between *NEFH* expression and the laminar distribution of cell types inferred by Pianno. This analysis revealed the highest correlation of *NEFH* expression with the distribution of L5 extratelencephalic (ET) neurons in M1C and ACC (Supplementary Fig. S5f, Fig. 6f), suggesting that *NEFH* tends to exhibit higher expression in L5 ET neurons, which have large size and long projections. This enrichment was not observed in snRNA-seq data is likely due to the low quantity of L5 ET in the human neocortex[45].

In summary, these results unveil the species-specific and region-specific presence of a subtype of neurons in deep L3 of the human neocortex, likely linked to the expansion of upper layers in the primate neocortex, particularly in regions associated with human- or

primate-specific functions[14]. These include the dlPFC, known for its involvement in high cognitive functions, and M1C, which likely evolved to accommodate fine motor control in humans[46,47]. The enrichment of *NEFH* and other neurofilament genes implies the expansion of long-range projection neurons in L3, potentially increasing the connections between neocortical areas and other telencephalic brain regions, thereby facilitating more complex information integration and processing.

These findings provide valuable insights into the evolution of cellular diversification and regional differences within the human neocortex, emphasizing the power of combining Pianno with unsupervised approaches and multimodal data analysis to reveal novel biological phenomena.

## Discussion

In this study, we demonstrated the remarkable performance of Pianno in semantic segmentation and annotation of diverse-shaped anatomical structures, as well as pathological foci and cell types, across data generated from various spatial technology platforms. Our study also showcased Pianno as a valuable tool for replacing labor-intensive manual annotation procedures and facilitating the revelation of novel biological insights, when integrated with unsupervised clustering methods.

The augmentation in the performance of Pianno can be attributed to the innovative treatment of marker genes as a pseudo-image within the pattern detector module, furnishing a robust prior distribution for the Bayesian classifier. This classifier seamlessly integrates the Markov random field (MRF) with the spatial Poisson point process (sPPP), leveraging sPPP's capability to model count data from RNA-seq while accounting for the covariance between spatially neighboring spots. In the subsequent MRF design, Pianno carefully considers both transcriptomic and spatial similarities, coupled with the global consistency between spots. This meticulous approach ensures the accurate refinement of labels, marking a key factor in Pianno's improved performance.

While Pianno has demonstrated remarkable power in spatial semantic annotation, it is imperative to acknowledge that its efficacy is inherently linked to the availability of well-defined initial markers and the existing molecular knowledge of the tissue, which may limit the algorithm's ability to uncover unknown biological patterns. Therefore, future endeavors should prioritize investigations that incorporate supplementary dimensions of information, such as cell size and density obtained by integrating hematoxylin and eosin (H&E) or 4',6-diamidino-2-phenylindole (DAPI) images. These augmentations could conceivably curtail Pianno's reliance on markers, thereby enhancing its overall robustness. Moreover, we envision a compelling avenue for advancing Pianno by integrating it with marker identification pipelines. This fusion presents an opportunity for Pianno to automatically refine its selection of markers. Consequently, it holds the potential to augment Pianno's ability to withstand noise in marker selection while reducing its dependency on prior knowledge. These advancements may collectively enhance the overall robustness and applicability of Pianno.

In conclusion, Pianno represents a powerful and efficient tool for spatial data analysis, and its potential can be further amplified by exploring marker refinement and the integration of complementary staining information. By continually advancing and refining the capabilities of Pianno, we can pave the way for more comprehensive and precise spatial analysis.

## Methods
### Ethical approval
Ethical approval for this study was obtained from the School of Basic Medical Sciences, Fudan University (Approval No. 2020-C006). Informed consent was obtained from the legal guardian of the donor.

## Data preparation

For spatial transcriptomics (ST) applications involving counts-based data from diverse platforms[2,3], Pianno leverages the gene-by-spot raw counts matrix in conjunction with two-dimensional spatial coordinates of each spot to initialize a Pianno object. This object is constructed atop the Anndata framework[25]. Subsequent to initialization, a two-fold data curation process unfolds. First, employing the SCANPY toolkit[25], low-quality genes detected in less than 1% of spots, as well as gene-sparse spots devoid of meaningful gene counts, are sieved out. Given the inherent sparsity of ST data, the application of SAVER[48] facilitates the derivation of a denoised gene expression matrix, which is subsequently subject to min-max normalization, thereby rescaling the denoised expression values. We suggest utilizing the *spatial_autocorr* function in Squidpy during preprocessing to retain the top 35% of genes with high spatial autocorrelation based on Moran's I Score.

The quality control is integrated in the *CreatePiannoObject* function of Pianno. Denoising can be achieved by calling the R package SAVER through Pianno. A Pianno object housing three pivotal data constituents:

1. $\mathbb{S} = \{s\}$: The set of spatial spot (location) captured in the region of interest (ROI). $|\mathbb{S}| = S, \forall s \in \mathbb{S}$ can be uniquely specified by its Cartesian coordinates $(x_s, y_s)$.
2. $C \in \mathbb{N}^{G \times S}$: The gene-by-spot raw counts matrix, wherein the raw expression vector of spot $s$ is denoted as $\boldsymbol{c}_s \in \mathbb{N}^G$.
3. $D \in \mathbb{R}^{G \times S}$: The denoised gene expression matrix, with each element $D_{gs} \in [0, 1]$ and the denoised expression vector of spot $s$ is denoted as $\boldsymbol{d}_s \in \mathbb{R}^G$.

## Pattern detector

Suppose there exists a total of R distinct spatial patterns to be discerned, with each pattern distinctly characterized by a collection of markers constituting a "marker list". Within this context, Pianno undertakes the transformation of the denoised marker gene expressions into pseudo-images, facilitating the initial label assignment for each spatial spot through the utilization of digital image processing techniques[49].

**Create binary mask**. An image is essentially a matrix. In addressing varying tissue coverage areas and spatial spots misalignment, a binary mask is created to assist in generating pseudo-images by establishing a mapping between spatial spots $\mathbb{S} \subset \mathbb{R}^2$ and image pixels $\mathbb{I}(\mathbb{S}) \subset \mathbb{Z}^2$ via a coordinate mapping function $f: (x_s, y_s) \mapsto (i, j)$ (see Supplementary Note). $\forall(i,j) \in \mathbb{I}(\mathbb{S})$ specified a pixel in row $i$, column $j$ on the image and correspond to at least one spatial spot located in the ROI.

The binary mask $M$ is initialized as an all-zero matrix with I rows and J columns, where I and J denote the maximum indexes of $i$ and $j$ in $\mathbb{I}(\mathbb{S})$ respectively. Then let $M_{ij} = 1$, if $(i,j) \in \mathbb{I}(\mathbb{S})$ and 0 otherwise to differentiate the ROI and background in the mask. The mask orchestrates the transformation of given spatial gene expression into a I × J digital image, on which the ROI's pixel value reflects average gene expression of its related spots, while the background remains steadfastly at "0".

**Create pattern image**. To generate the image representation for the $r$th pattern ($r \in \{1, 2, ..., R\}$) defined by G markers (G ≥ 1), a series of steps are performed. The outcome is a grayscale image denoted as $P^{(r)}$, with dimensions I × J.

Initially, the denoised gene expression values of the G markers are fused together using the median value. Notably, the background values remain at 0. Consequently, the ensuing discussion solely pertains to the pixel values within the ROI. i.e. $\forall(i,j) \in \mathbb{I}(\mathbb{S}), \exists \mathbb{S}^* \subset \mathbb{S}, \mathbb{S}^* \neq \varnothing$, s.t.

$$P_{ij}^{(r)} = \frac{1}{|\mathbb{S}^*|} \sum_{\forall s \in \mathbb{S}^*} \text{Median}\{\boldsymbol{d}_s\}. \tag{1}$$

Subsequently, the scikit-image[50] is harnessed for the purpose of conducting median filtering on $P^{(r)}$. The specific operation is performed using the *filter.median* function, implemented with a 3 × 3 kernel. Mathematically, the process can be expressed as follows: $\forall(i,j) \in \mathbb{I}(\mathbb{S})$,

$$P_{ij}^{(r)} = \text{Median}\{P_{ij}^{(r)}, P_{i\pm1 j\pm1}^{(r)}\}. \tag{2}$$

The resulting image $P^{(r)}$ is then utilized as the $r$th pattern channel, ultimately yielding a multichannel pattern image denoted as $P$. Pattern image $P$ possesses dimensions I × J × R, with individual elements given by $P_{ijr} = P_{ij}^{(r)}$.

**Mask-based feature extraction**. The subsequent phase involves the extraction of features from the pattern image $P$, aimed at the identification of each target pattern. Within the context of the $r$th pattern channel $P^{(r)}$, a segmentation procedure is enacted, subdividing $P^{(r)}$ into $n$ ($n$ is set to 3 as default, representing high, medium and low gene expression levels) distinct regions. This segmentation is achieved through Multi-Otsu thresholding[51]. The process yields $n - 1$ thresholds denoted as $\tau_1 > \tau_2 > \cdots > \tau_{n-1}$, effectively partitioning pixel intensities into $n$ levels.

The region within $P^{(r)}$ characterized by the highest intensity level corresponds to the most probable location of the $r$th pattern, thus designated as the positive image denoted $P^{(r)+}$. This can be mathematically represented as: $\forall(i,j) \in \mathbb{I}(\mathbb{S})$

$$P_{ij}^{(r)+} = \begin{cases} P_{ijr}, & P_{ijr} > \tau_1 \\ 0, & P_{ijr} \leq \tau_1. \end{cases} \tag{3}$$

The complementary image, termed $P^{(r)-}$, is defined as follows:

$$P_{ij}^{(r)-} = \begin{cases} 0, & P_{ijr} > \tau_1 \\ P_{ijr}, & P_{ijr} \leq \tau_1. \end{cases} \tag{4}$$

Given the potential influence of noise on $P^{(r)+}$, the emergence of false positive responses or minor discontinuities is plausible. To address this, the subsequent steps are undertaken:

1. **Identification of Connected Components in $P^{(r)+}$**: Connected components within $P^{(r)+}$ are labeled. Subsequently, small components with an area less than 2 (user-specified) are filtered out. Pixels are connected when they are neighbors in an 8-connected sense and neither has a value of 0.
2. **Denoising of $P^{(r)+}$**: The *denoise_tv_chambolle* function from the scikit-image library is employed for denoising $P^{(r)+}$. Notably, this step involves the calculation of the average pixel intensity $p_k^+$ for each positive component $k$ in $P^{(r)+}$, along with the average pixel intensity $n^-$ of a negative component drawn from $P^{(r)-}$. Subsequently, K-means clustering with K = 2 is performed on the set $\mathbb{C} = n^- \cup \{p_k^+\}_{k \geq 1}$, incorporating the negative component to enhance the detection of false positive responses. This strategy ensures the robustness of 2-means clustering, even in scenarios involving only one positive connected component within $P^{(r)+}$. The outcome of the clustering operation classifies connected components within $P^{(r)+}$ into positive and negative categories based on their respective clustering centers. Consequently, components belonging to the negative class are removed from $P^{(r)+}$, serving to eliminate false positive responses.
3. **Binarization and Dilation[49]**: Binarization of $P^{(r)+}$ generates a binary mask denoted as $M^{(r)+}$ for the $r$th pattern. Subsequently, dilation is applied to expand the positive region and bridge minor gaps, facilitated by a user-defined dilation radius (to be optimized, default value is 2).
4. **Feature Extraction**: The $r$th feature $F^{(r)}$ is extracted from $P^{(r)}$ by employing $M^{(r)+}$ as a mask. Mathematically, $\forall(i,j) \in \mathbb{I}(\mathbb{S})$, the

process is represented as:

$$F_{ij}^{(r)} = P_{ijr} \times M_{ij}^{(r)+}. \tag{5}$$

5. **Post-Processing of Features**: The extracted feature $F^{(r)}$ undergoes a sequence of operations, including sharpening, Gaussian smoothing, and noise reduction[49,50]. Subsequently, the intensity of $F^{(r)}$ is rescaled to fall within the range $[0, 1]$.

Collectively, the $r$th feature channel is established as $F^{(r)}$, thereby culminating in the creation of a composite feature image $F$ with dimensions $I \times J \times R$, where individual elements are denoted by $F_{ijr} = F_{ij}^{(r)}$.

**Label initialization.** The initialization of labels involves the assignment of initial patterns to spatial spots, facilitated by the representation of a spatial spot $s$ using an R-dimensional feature vector denoted as $\boldsymbol{f}_s$. Each component within this vector ($r$th component) measures the likelihood of spot $s$ being associated with the $r$th pattern.

Given the correspondence between spatial spots and pixels within the feature image $F$, $\forall s \in \mathbb{S}, \exists!(i,j) \in \mathbb{I}(\mathbb{S})$, s.t.

$$\boldsymbol{f}_s = \left[ F_{ij1}, F_{ij2}, \cdots, F_{ijR} \right]^\top. \tag{6}$$

The initial label $l_s$ for each spot $s$ is determined by selecting the pattern with the highest score on the corresponding feature vector $\boldsymbol{f}_s$. Mathematically, this process can be expressed as:

$$l_s = \arg\max_r \{\boldsymbol{f}_s\} = r^*, \quad r^* \in \{1, 2, \ldots, R.\} \tag{7}$$

In certain scenarios, the exact number of spatial patterns present in the tissue may be unknown, or only specific patterns are of interest. Additionally, instances might arise where information regarding a pattern's marker is unavailable. To accommodate such cases, Pianno offers the flexibility to incorporate an undefined pattern, characterized by an uncertainty parameter $u \in [0, 1]$ (default is 0.5). The corresponding feature $F^{(u)}$ for this undefined pattern is calculated as follows:

$$F_{ij}^{(u)} = u - \frac{1}{R}\sum_{r=1}^{R} F_{ijr}. \tag{8}$$

By merging this undefined feature with the remaining R features, a $I \times J \times (R+1)$ feature image $F$ is constructed, featuring an additional channel for the undefined pattern. This undefined pattern can be designated as "Background" and Pianno accommodates the option to disregard it, considering it as part of the background.

**Automated pattern detection and marker selection.** The pattern detection process necessitates the provision of markers corresponding to patterns to be identified, with each image processing step–such as Multi-Otsu thresholding, dilation, sharpening, Gaussian blurring, and noise reduction–requiring configuration of relevant parameters. To enhance user-friendliness, we employ the NNI (Neural Network Intelligence)[52] toolkit for automating hyperparameter optimization. Users are only required to specify as few as one known marker for each pattern as a priori information, with up to one pattern allowed to have no initial marker.

Assuming that the set of all parameters to be optimized is $\mathbb{H}$ and the search space is $\Omega$, for the $t$th trial, the tuner (default is TPE) selects a parameter set $\mathbb{H}^{(t)}$ from the search space according to the sampling strategy. Then pattern detector can label the spatial spots with patterns based on the initial markers and $\mathbb{H}^{(t)}$. Then we can compute the average expression in each type of spots for each initial marker and perform min-max normalization by category to obtain the specificity

matrix $T(\mathbb{H}^{(t)}) \in [0,1]^{R \times R}$ of the initial marker on the current annotation. The closer $T(\mathbb{H}^{(t)})$ is to the identity matrix $I_R$, the better the current annotation matches the known information, and thus the most suitable parameter set $\mathbb{H}^{(t)}$ is automatically selected. The objective function of the optimization is defined as the Euclidean distance between the specificity matrix $T(\mathbb{H}^{(t)})$ and the identity matrix $I$:

$$\min d^{(t)} = \parallel T(\mathbb{H}^{(t)}) - I_R \parallel \tag{9}$$

NNI facilitates the identification of the optimal parameter set within a reasonable timeframe, achieving the automatic initialization of annotation. Furthermore, Pianno generates additional candidate markers based on the initial annotation through the Wilcoxon rank-sum test in SCANPY. Users can then expand the marker list through selection from the candidate list. The configuration of all experimental parameters and the list of initial and updated marker genes in this paper are detailed in GitHub project of Pianno.

**Bayesian classifier**
The initial annotation is obtained through pattern detection. However, many operations such as noise reduction, smoothing, and sharpening during image processing may destroy the original biological features. Therefore, a Bayesian classifier is built based on the raw counts to fine-tune the initial annotation. To annotate spots into patterns, we calculate the posterior probability $p(l_s = r | \boldsymbol{c}_s, \hat{\Theta})$ that each spot $s$ is of a given pattern $r$, where $\hat{\Theta}$ is the maximum a posterior (MAP) of model parameters.

**High-order MRF prior model.** The spot-type prior $p(l_s = r)$ is characterized through a high-order Markov random field (MRF) model based on the initial annotations[53]. $\forall s \in \mathbb{S}$, the neighborhood $\mathcal{N}(s) = \mathcal{N}_U(s) \cup \mathcal{N}_S(s)$ is divided into two components: the K-nearest neighbors $\mathcal{N}_U(s)$ in the UMAP space determined by transcriptomic similarity, and the spatial neighbors $\mathcal{N}_S(s)$ determined by the spatial proximity.

To quantify the cost of assigning label $r$ to spot $s$ concerning local similarity, the unary term $\Phi(l_s = r)$ is formulated as:

$$\Phi(l_s = r) = -[\boldsymbol{\omega}_1 \ln p(l_s = r | \mathcal{N}_U(s)) + (1 - \boldsymbol{\omega}_1)\ln p(l_s = r | \mathcal{N}_S(s))], \tag{10}$$

where $\boldsymbol{\omega}_1 \in [0, 1]$ is a hyperparameter. Here, $p(l_s = r | \mathcal{N}_U(s))$ and $p(l_s = r | \mathcal{N}_S(s))$ represent the frequency of spots labeled $r$ within $\mathcal{N}_U(s)$ and $\mathcal{N}_S(s)$, respectively. The term $\Phi(l_s = r)$ takes into account both transcriptomic similarity and spatial proximity, acknowledging that spots with similar gene expressions might share the same label even if they are spatially distant.

To assess the cost of assigning label pairs $(r, r')$ to spot pairs $(s, s')$ from a global perspective, a pairwise term $\Psi(l_s = r, l_{s'} = r')$ is established:

$$\Psi(l_s = r, l_{s'} = r') = -\mathcal{D}(\boldsymbol{f}_s, \boldsymbol{f}_{s'}) \times \ln p_{rr'}, \tag{11}$$

Here, $p_{rr'}$ signifies the co-occurrence rate of label $r$ and $r'$, computed by tallying the frequency of label pairs $(r, r')$ from neighboring spots $s$ and $s'$. If $p_{rr'}$ is lower in the whole initial annotation, then the cost of assigning the label of spot $s$ whose neighborhood label is $r'$ to $r$ will be higher. The penalty term $\mathcal{D}(\boldsymbol{f}_s, \boldsymbol{f}_{s'})$ takes the Euclidean distance between the feature vectors $\boldsymbol{f}_s$ and $\boldsymbol{f}_{s'}$ into account:

$$\mathcal{D}(\boldsymbol{f}_s, \boldsymbol{f}_{s'}) = \mathbf{I}_{\{r=r'\}} \cdot e^{\|\boldsymbol{f}_s - \boldsymbol{f}_{s'}\|_2} + (1 - \mathbf{I}_{\{r=r'\}}) \cdot (1 + e^{-\|\boldsymbol{f}_s - \boldsymbol{f}_{s'}\|_2}), \tag{12}$$

In this equation, $\mathbf{I}\{r = r'\}$ is an indicator function that equals 1 when $r = r'$ and 0 otherwise. If two neighboring spots look similar on the feature image ($\| \boldsymbol{f}_s - \boldsymbol{f}_{s'} \|_2 \approx 0$), then assigning different labels to them will be penalized more heavily ($\mathcal{D}(\boldsymbol{f}_s, \boldsymbol{f}_{s'}) \approx 2$) than assigning the same label ($\mathcal{D}(\boldsymbol{f}_s, \boldsymbol{f}_{s'}) \approx 1$). If two neighboring spots look different on the

feature image ($\|\boldsymbol{f}_s - \boldsymbol{f}_{s'}\|_2 \gg 0$), then assigning same labels to them will be penalized more heavily ($\mathcal{D}(\boldsymbol{f}_s, \boldsymbol{f}_{s'}) \gg 1$) than assigning the different labels ($\mathcal{D}(\boldsymbol{f}_s, \boldsymbol{f}_{s'}) \approx 1$).

Introducing the hyperparameter $\boldsymbol{\omega}_2 \in [0,1]$ as the weight of $\Psi(l_s = r, l_{s'} = r')$, the high-order energy function $E_{sr}$ is obtained by combining the unary and pairwise terms:

$$E_{sr} = (1 - \boldsymbol{\omega}_2)\Phi(l_s = r) + \boldsymbol{\omega}_2 \sum_{\forall s' \in \mathcal{N}(s)} \Psi(l_s = r, l_{s'} = r'). \tag{13}$$

Notably, in order to make sense of the weighted summation (Eq. (13)), each component is scaled to the range [0,1] using Min-Max normalization. Therefore, $\forall s \in \mathbb{S}, E_{sr} \in [0,1]$. Consequently, the MRF prior model can be expressed as a Gibbs distribution[54]:

$$\boldsymbol{\pi}_{sr} = \frac{e^{\kappa \times E_{sr}}}{\sum_{r=1}^{R} e^{\kappa \times E_{sr}}}. \tag{14}$$

Here, $\kappa$ serves as a scale factor (default is 3) to adjust the strength of the prior. $\forall s \in \mathbb{S}, \boldsymbol{\pi}_s = (\boldsymbol{\pi}_{s1}, \boldsymbol{\pi}_{s2}, \cdots, \boldsymbol{\pi}_{sR})$. By integrating Equations (10) to (14), the spot-type prior $p(l_s = r | \boldsymbol{\pi}_s) = \boldsymbol{\pi}_{sr}$ can be initialized using specific $\boldsymbol{\omega}_1$ and $\boldsymbol{\omega}_2$ values based on the initial annotations generated by the pattern detector.

**Spatial Poisson point process model.** Existing research demonstrates that the negative binomial distribution can aptly model the raw counts from single-cell sequencing[24,48]. However, in the context of spatial transcriptomics, the gene expression $C_{gs}$ and spatial location $(x_s, y_s)$ of each spot $s$ must be jointly considered. Therefore, we adopt a homogeneous spatial Poisson point process (sPPP) model for the $C_{gs} | l_s = r$ distribution[55,56]:

$$C_{gs} | l_s = r \sim \text{Poisson}(\sigma_s \cdot \boldsymbol{\lambda}_{gsr}). \tag{15}$$

Here, $\sigma_s$ represents the size factor of spot $s$, calculated using scran[57]. The intensity function $\lambda_{gsr}$ of the sPPP denotes the average expression count of gene $g$ per spot and varies with spatial location $(x_s, y_s)$.

By establishing a connection between the Poisson distribution and the negative binomial distribution, where if a random variable $x \sim \text{Poisson}(k\lambda)$ and $\lambda \sim \text{Gamma}(\alpha, \frac{\alpha}{\mu})$, then $x \sim \text{NB}(\alpha, k\mu)$, we can assume that the intensity function $\lambda_{gsr}$ follows a Gamma distribution:

$$\boldsymbol{\lambda}_{gsr} \sim \text{Gamma}\left(\alpha_{gsr}, \frac{\alpha_{gsr}}{\mu_{gsr}}\right). \tag{16}$$

In this equation, $\alpha_{gsr}$ is the shape parameter of the Gamma distribution, and $\mu_{gsr}$ is the mean of the intensity function $\lambda_{gsr}$. This leads to:

$$C_{gs} | l_s = r \sim \text{NB}(\alpha_{gsr}, \sigma_s \times \mu_{gsr}). \tag{17}$$

Consequently, the probability distribution of $C_{gs}$ given $l_s = r$ is:

$$\begin{aligned} p(C_{gs} | l_s = r) &= \text{NB}(C_{gs}; \alpha_{gsr}, \sigma_s \times \mu_{gsr}) \\ &= \frac{\Gamma(C_{gs} + \alpha_{gsr})}{\Gamma(\alpha_{gsr})\Gamma(C_{gs} + 1)} \left(\frac{\alpha_{gsr}}{\alpha_{gsr} + \sigma_s \times \mu_{gsr}}\right)^{\alpha_{gsr}} \left(\frac{\sigma_s \times \mu_{gsr}}{\alpha_{gsr} + \sigma_s \times \mu_{gsr}}\right)^{C_{gs}}. \end{aligned} \tag{18}$$

Here, $\alpha_{gsr}$ and $\sigma_s \times \mu_{gsr}$ correspond to the inverse dispersion and the mean of the negative binomial distribution, respectively. The inverse dispersion $\alpha_{gsr}$ can be calculated based on $\sigma_s \times \mu_{gsr}$[24,58]:

$$\alpha_{gsr} = \sum_{i=1}^{B} \boldsymbol{a}_i \times \exp\{-b \times (\sigma_s \times \mu_{gsr} - x_i)^2\}, \tag{19}$$

where $\boldsymbol{a}_i$ represents the parameters of the radial basis functions to be estimated. B signifies the total number of radial basis function centers, and $x_i$ is the center $i$. These centers are evenly spaced from 0 to $\max\{C_{gs}\}$. The constant $b$ is set as twice the square difference between the first and second centers.

To parameterize the negative binomial distribution, Pianno defines the mean $\mu_{gsr}$ of the intensity function in sPPP as follows:

$$\mu_{gsr} = \boldsymbol{\beta}_{gs} + \beta_{gr}\rho_{gr}\boldsymbol{\delta}_{gr}. \tag{20}$$

In this equation, $\boldsymbol{\beta}_{gs}$ signifies the baseline expression of gene $g$ in spot $s$, estimated to capture the influence of spatial location on gene expression. $\beta_{gr}$ represents the mean of the raw counts of gene $g$ within an area encompassing all spots belonging to pattern $r$. $\beta_{gr}$ corresponds to the average raw counts of gene $g$ on the skeleton of pattern $r$, which characterizes the impact of spot biological identity on gene expression. The skeleton is derived using the *morphology.skeletonize* function in scikit-image, representing the central locations of a pattern[49,49,50]. Thus, spots located on the skeleton hold a higher confidence in representing a pattern compared to other spots. All input genes are derived from the marker list. An indicator variable $\rho_{gr}$ indicates the binary relationship between gene $g$ and pattern $r$. When gene $g$ is a marker of pattern $r$, $\rho_{gr}$ equals 1; otherwise, it's 0. The spot-pattern-specific overexpression term $\boldsymbol{\delta}_{gr} > 1$ is introduced. It indicates that the expression of gene $g$ in spots of pattern $r$ is above average when gene $g$ is a marker of pattern $r$, while it fluctuates primarily with spatial location otherwise.

## Inference

To optimize the introduced hidden variable $\boldsymbol{l} = \{l_s\}$, Pianno employs the Expectation-Maximization (EM) algorithm to refine its parameter space $\Theta = \{\boldsymbol{\omega}, \boldsymbol{\beta}, \boldsymbol{\delta}, \boldsymbol{a}, \boldsymbol{\pi}\}$, where $\boldsymbol{\omega} = \{\boldsymbol{\omega}_1, \boldsymbol{\omega}_2\}$, $\boldsymbol{\beta} = \{\boldsymbol{\beta}_{gs}\}$, $\boldsymbol{\delta} = \{\boldsymbol{\delta}_{gr}\}$, $\boldsymbol{a} = \{\boldsymbol{a}_i\}$, and $\boldsymbol{\pi} = \{\boldsymbol{\pi}_s\}$. The spot-pattern prediction for each spot is facilitated by the conditional posterior $p(l_s = r | \boldsymbol{c}_s, \hat{\Theta})$, under the assumption that the raw counts of different genes on each spot are independent and identically distributed (i.i.d.). This posterior probability is denoted as $p_{sr}$:

$$p(l_s = r | \boldsymbol{c}_s, \hat{\Theta}) = \frac{p(l_s = r | \boldsymbol{\pi}_s) \prod_g p(C_{gs} | l_s = r)}{\sum_{r'} p(l_s = r' | \boldsymbol{\pi}_s) \prod_g p(C_{gs} | l_s = r')} \triangleq p_{sr} \tag{21}$$

**Parameter initialization.** Pianno initializes its parameters as follows:
1. Both initial $\boldsymbol{\omega}_1 \in [0,1]$ and $\boldsymbol{\omega}_2 \in [0,1]$ are user-specified (default values are 0.5).
2. Dirichlet$(0.01, 0.01, \cdots, 0.01)$ is used as a hyper-prior on $\boldsymbol{\pi}_s$[24], and $\boldsymbol{\pi}_s$ is initialized using the MRF prior model (Eq. (10) - Eq. (14)).
3. $\boldsymbol{\beta}_{gs}$ is randomly initialized using a draw from $\mathcal{N}(0,1)$.
4. $\boldsymbol{a}_i$ is initialized to 1.

**E step.** The log-likelihood of the complete-data is calculated as:

$$L(\Theta) = \sum_{g=1}^{G} \log[p(C_{gs} | l_s = r, \Theta)p(l_s = r | \boldsymbol{\pi}_s)p(\boldsymbol{\pi}_s)]. \tag{22}$$

Let $\Theta^{(t)}$ be the parameter estimate at iteration $t$. Let $p_{sr}^{(t)}$ represent the conditional posterior $p(l_s = r | \boldsymbol{c}_s, \Theta^{(t)})$. The $Q$ function is then defined as:

$$Q(\Theta, \Theta^{(t)}) = \sum_{s=1}^{S} \sum_{r=1}^{R} p_{sr}^{(t)} \sum_{g=1}^{G} \log[p(C_{gs} | l_s = r, \Theta)p(l_s = r | \boldsymbol{\pi}_s)p(\boldsymbol{\pi}_s)]. \tag{23}$$

**M step.** The goal is to maximize the $Q$ function to determine the updated parameter estimate $\Theta^{(t+1)}$ at the $(t+1)$-th iteration:

$$\Theta^{(t+1)} = \arg\max_{\Theta} Q(\Theta, \Theta^{(t)}). \tag{24}$$

The parameters of the $Q$ function are optimized using the Adam optimizer in the M-step, with a default learning rate of 0.1 and a maximum of $10^5$ iterations. The iteration stops when $\|Q(\Theta^{(t+1)},\Theta^{(t)})-Q(\Theta^{(t)},\Theta^{(t-1)})\|<10^4$, and the final parameter estimate is denoted as $\hat{\Theta}=\Theta^{(t+1)}$.

**Label renewal.** There are two methods for renewing the labels. The first method is to assign the label of a spot $s$ as the pattern with the highest posterior probability (method = "argmax"), i.e.,

$$l_s = \arg\max_r p(l_s = r|\boldsymbol{c}_s,\hat{\Theta}). \qquad (25)$$

The second method, referred to as "imgbase", treats the obtained posterior probability distribution as an optimized pattern image and uses it to update the annotation from the pattern detector. These two methods are suitable for different scenarios. The "argmax" method is recommended for cell type annotation since it can capture subtle features, while the "imgbase" method is better suited for structural annotation as it maintains the continuity of the structure.

**Sensitivity analysis**
We designed three control variable experiments using the dlPFC dataset to investigate the impacts of (a) the number of marker genes; (b) the number of thresholds of the Multi-Otsu thresholding algorithm; and (c) $\boldsymbol{\omega}_1$ and $\boldsymbol{\omega}_2$ of the High-Order MRF Prior Model on Pianno's performance.

(a) We first constructed a total marker list of 10 markers per pattern that appear in the top 10 candidate marker genes for at least one sample. For each trial, we fixed the other parameters, while randomly selected N (N increments from 1 to 10) genes from the total marker list. Subsequently, we evaluated Pianno's performance by varying the number of marker genes per pattern randomly selected from the total marker list. Notably, we observed an improvement in Pianno's accuracy with an increasing number of markers, surpassing a commendable accuracy (ACC > 0.6) when the number of markers per pattern exceeded 50% (Supplementary Fig. S6a). This trend is likely attributed to the inclusion of noise marker genes in each sample, where certain genes may rank among the top marker genes in one sample but fail to exhibit pattern-specific expression in other samples due to technical variations like dropouts.

To further explore the impact of noise inclusion in the marker list on Pianno's accuracy, we calculated the percentage of noise genes–those not included in the top 10 marker list of a given sample when using the total marker list (Supplementary Fig. S6b). The average percentage of noise genes was found to be 47%. Thus when the number of markers is more than 50%, the mean value of ACC can be stable above 0.6. This suggests that the inclusion of randomly selected noise genes may be compensated for by the increased number of genes, presenting a higher chance to include high-quality markers. In contrast, selecting a small number (1-3) of personalized markers for each sample resulted in an enhanced accuracy of Pianno (Supplementary Fig. S6b). Overall, these results underscore the importance of the inclusion of high-quality markers, which can be achieved by either increasing the number of top marker genes with potential noise or by adopting individualized marker selection for each sample starting with minimal prior information.

(b) While maintaining other parameters unchanged and utilizing a manually curated marker list, the performance of Pianno was evaluated by systematically increasing the values of thresholds in the Multi-Otsu thresholding algorithm from 1 to 5. The accuracy of Pianno's annotations for each sample was then calculated. Notably, when $n=1$, no thresholding segmentation is performed, and for excessively large n values ($n>5$), segmentation was impractical due to restricted gene expression value ranges. The optimal performance of Pianno was observed when n equaled 2 or 3, reflecting the states of presence and absence or categorization into high, medium, and low gene expression levels, respectively (Supplementary Fig. S6c).

(c) While keeping other parameters constant, the performance of Pianno was assessed by increasing the values of $\boldsymbol{\omega}_1$ and $\boldsymbol{\omega}_2$ from 0 to 1 with a step size of 0.01. This was done to calculate the accuracy of Pianno's annotation specifically for sample 151673. The hyperparameter $\boldsymbol{\omega}_1$ represents the effects of transcriptome and spatial similarity, while $\boldsymbol{\omega}_2$ represents the global consistency of the prior distribution. The experimental results indicated that a larger $\boldsymbol{\omega}_1$ combined with a smaller $\boldsymbol{\omega}_2$ achieved the best annotation results (Supplementary Fig. S6d). It's noteworthy that, in studies in this manuscript, we default to using $\boldsymbol{\omega}_1 = 0.99$ and $\boldsymbol{\omega}_2 = 0.01$

**Benchmark analysis of layer structure annotation**
In the benchmark analysis of layer structure annotation for the human dlPFC[16], multiple other spatial clustering methods were employed. These methods utilized the same number of clusters (7) as the actual layers, following the parameter settings recommended by the author in the original paper or tutorial for each method (CellAssign[24], SCANPY[25], SpaGCN[8], SEDR[9], BayesSpace[7], DeepST[10], and STAGATE[11]). We used the data with and without preprocessing by SAVER as input, respectively. The performance of these methods was evaluated using the Adjusted Rand Index (ARI), which measures the similarity between the spatial domains (i.e. regions with similar spatial expression patterns) obtained by these methods and the manual annotation.

To facilitate a comprehensive comparison of Pianno's performance with other spatial clustering methods, the spatial domain labels obtained from these methods were mapped to the manually annotated structural labels using the Kuhn-Munkres algorithm[59]. This mapping ensured a fair comparison of the results. Various classification metrics were then computed to assess the performance of different methods, including accuracy (ACC), macro-averaging precision (P), macro-averaging recall (R), macro-averaging F1-score (F1), and normalized mutual information (NMI). These metrics provided a multifaceted evaluation of how well each method aligned with the manual annotation.

**Differentially expressed analysis**
We used the Wilcoxon rank-sum test implemented in SCANPY[25] to identify differentially expressed genes for each cluster/cell-type/structure with default parameters.

**Cell-type deconvolution of VISp**
We compared the cell-type deconvolution results of three methods:[18], Tangram[17], and RCTD[33], with the results obtained by Pianno using spatial transcriptome data from the mouse primary visual cortex (VISp). The cell-type marker genes utilized by Pianno were derived from the differentially expressed analysis of scRNA-seq data, employing the default parameter settings.[18], Tangram[17], and RCTD[33] were executed with the parameter settings recommended by the authors in their original papers or official tutorials.

**Assign clusters to structures**
After Pianno's structure annotation, we calculated the proportion of unsupervised clusters within each structure. By assigning clusters to the structure with the highest proportion, we gained a better understanding of the distribution and composition of unsupervised clusters within different anatomical structures. This information contributes to our insights into the complexity and organization of tissue structures.

**Cell-type co-occurrence analysis**
We computed and visualized co-occurrence probability of cell-type clusters in the breast cancer samples through the co-occurrence function implemented in Squidpy[60] with default parameters.

## Gene ontology (GO) enrichment analysis

We conducted Gene Ontology (GO) enrichment analysis using the top 100 differentially expressed genes of cluster A-11 in the human dlPFC sample 151671. This analysis was performed by the "Functional Annotation Tool" implemented in the database for annotation, visualization and integrated discovery (DAVID, https://david.ncifcrf.gov/). The raw P-values are estimated by DAVID website using one-sided Fisher's exact test without adjustment.

## Spatial transcriptome data generation and analysis

**Experimental model and subject details.** To obtain spatial transcriptome data, the primary motor cortex (M1C) and anterior cingulate cortex (ACC) samples were obtained from a healthy adult human brain specimen (female, 73 years old) at the Fudan University Body Donation Receiving Station of Shanghai Red Cross, under the approval from the ethics committee at the School of Basic Medical Sciences, Fudan University (Approval No. 2020-C006).

**Tissue processing and Visium data generation.** Tissue RNA integrity (RIN) was assessed using Bioanalyzer 2100 (Agilent), while tissue morphology was examined through hematoxylin (51275, Sigma-Aldrich) and eosin (109844, Sigma-Aldrich) (H&E) staining. All procedures were carried out in accordance with the instructions outlined in the Methanol Fixation, H&E Staining & Imaging for Visium Spatial Protocols provided by 10× Genomics. Tissues that exhibited RIN values above 7.0 and displayed a well-organized structure were selected for subsequent procedure. To prepare the tissue for sectioning, Tissue-Tek® O.C.T. Compound (4583, SAKURA) was applied onto dry ice for embedding, and the tissues were then stored at -80 °C until further processing. For brain samples, they were thawed from -80 °C storage and allowed to stabilize at -20 °C in a cryostat (CM1950, Leica) for 30 min before sectioning. Subsequently, sections with a thickness of 10 μm were collected.

To prepare the libraries for sequencing, we utilized the Tissue Optimization and Spatial Gene Expression Kits from Visium Spatial Gene Expression Slide & Reagent Kit (1000184, 10x Genomics). All procedures were performed in accordance with the manufacturer's instructions. Briefly, brain tissue section underwent initial processing through H&E staining and imaging. Subsequently, tissue permeabilization was conducted to capture mRNA. This was followed by reverse transcription, second strand synthesis, cDNA amplification, fragmentation, and final library amplification. The resulting libraries were then sequenced on a NovaSeq6000 platform, targeting approximately 400 million reads from tissue-specific spots using paired-end, dual indexed sequencing. The run parameters consisted of a read1 length of 150 bp, an i7 index length of 10 bp, an i5 index length of 10 bp, and a read2 length of 150 bp.

**Downstream analysis.** After raw data preprocessing, the layer structures of M1C and ACC were initially annotated using Pianno. Subsequently, the gene expression matrices of M1C and ACC were integrated based on the top 2000 highly variable genes using the scanorama method[61]. The layer labels assigned by Pianno for these two brain regions were successfully combined on the UMAP plot, indicating the effectiveness of the integration process.

Afterwards, the integrated gene expression matrix was subjected to clustering using the Leiden algorithm, implemented in SCANPY[25] with default parameters (resolution=1). The goal was to identify distinct clusters and assess differences in their composition between M1C and ACC. Then differential expression analysis was performed on cluster B-4, which was exclusive to the M1C deep layer 3.

## Statistics & reproducibility

No statistical method was used to predetermine sample size. The number of samples were chosen for this exploratory study based on the availability of materials at study time. The spatial transcriptome data generated in this study was isolated from a single donor as a validation. We did not include data that were clearly outliers in spatial transcriptome in the analysis. Low-quality genes detected in less than 1% of spots, as well as spots without gene counts detected, are sieved out. All experimental steps are detailed in the methods to ensure replication. A reproducible tutorial for each experiment is accessible at https://pianno-tutorials.readthedocs.io/en/latest/index.html. The study is exploratory and descriptive to demonstrate the utility of a computational method, and no case control comparisons were performed, so no randomization was considered. No blinding as no case-control comparisons are made.

## Reporting summary

Further information on research design is available in the Nature Portfolio Reporting Summary linked to this article.

## Data availability

All relevant data supporting the key findings of this study are available within the article and its Supplementary Information files. The raw spatial transcriptome data of human M1C and ACC generated in this study have been deposited in the Genome Sequence Archive (GSA) under accession code HRA004425. The raw data are available under controlled access for the nature of human genomics data, can be requested through the GSA platform. Additionally, the processed spatial transcriptome data are available at https://github.com/yuqiuzhou/Pianno[62].

All public datasets utilized in this study are accessible in their raw form from the respective original authors. Specifically, the human dlPFC dataset[16] is available within the spatialLIBD[63] (https://research.libd.org/spatialLIBD/). The processed Stereo-seq datasets from adult mouse coronal hemibrain and olfactory bulb are available at the Spatial Transcript Omics DataBase (STOmics DB) (https://db.cngb.org/stomics). The pre-processed Slide-seqV2 dataset from mouse hippocampus[37] is accessible within the Squidpy package (https://github.com/scverse/squidpy). The ST datasets of human pancreatic ductal adenocarcinoma[39] are available at the Gene Expression Omnibus under accession number GSE111672. The Visium datasets of human breast cancer are collected from the 10× Genomics website (https://support.10xgenomics.com/spatial-gene-expression/datasets). The scRNA-seq dataset from mouse primary visual cortex is available at the NCBI Gene Expression Omnibus (GEO) under accession GSE115746. The snRNA-seq dataset from multiple human cortical areas is available at Allen Brain Map (https://portal.brain-map.org/). The DAPI staining image of mouse olfactory bulb is accessible on https://github.com/JinmiaoChenLab/SEDR_analyses. The configuration of all experimental parameters and the list of initial and updated marker genes used in this paper are available at https://github.com/yuqiuzhou/Pianno[62].

Source data are provided in this paper[64].

## Code availability

Pianno is available as a Python package at https://github.com/yuqiuzhou/Pianno[62].

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

## Acknowledgements

We thank S.Z. and P.L. for providing advice on the mathematical expression improvement. We also thank the Fudan University Body Donation Receiving Station of Shanghai Red Cross for providing postmortem tissue. This work was supported by grants from the National Key R&D Program of China (Grant No. 2023YFF1204802), the STI2030-Major Projects (Grant No. 021ZD0200100 to Y.Z.), the National Natural Science Foundation of China (Grant No. 82071259 to Y.Z.), the Shanghai Municipal Science and Technology Major Project (Grant No. 2018SHZDZX01), ZJ Lab and Shanghai Center for Brain Science and Brain-Inspired Technology, China.

## Author contributions

Y.Z. and Y.Q.Z. conceived the study. Y.Q.Z. designed the algorithm and developed the Pianno python package. Y.Q.Z. and W.H. conducted computational experiments and model validation. W.Z.H. conducted spatial transcriptome experiments and raw data processing. W.Z.H. contributed to data collection. Y.Z. supervised the work. Y.Q.Z. and Y.Z. wrote the manuscript with approval of all authors.

## Competing interests

The authors declare no competing interests.
