## [Peer Review File · Nature Communications]

Pianno: a probabilistic framework automating semantic annotation for spatial transcriptomicsReviewer #1 (Remarks to the Author):

In this manuscript, the authors developed a new computational model, Pianno, to identify structural semantics based on marker genes. The authors showed that Pianno can estimate cell type distributions and annotate a wide array of spatial semantics. Actually, the pattern image generated by the pattern detector module of Pianno is a linear combination of the denoised expression of marker genes, which makes the selection of marker genes particularly important. However, how to select the marker gene list and parameter settings are not described in the paper, and its robustness and scalability need to be further validated.

Major concerns:

1. The selection process for marker genes should be described in detail and avoid manual selection. The core step of Pianno is to obtain the pattern image by linearly fusing the expression of marker genes. But how to select the marker gene list is not described in the paper. Authors should ensure that this step is fully automated.
2. Is Pianno sensitive to the choice of marker gene list? Will performance degrade if 10% or 20% of the genes in the marker gene list are randomly selected?
3. Pianno stores gene expression in the form of images (i.e., dense matrix). Can Pianno handle large-scale data? If scale factor is introduced, is it equivalent to reducing the spatial resolution of the sequencing technology?
4. The algorithm may potentially be affected by "p-hacking", where parameters/settings are tuned to get the best performance of the reported data. As the study did not use many datasets, such a good performance may not be transferable to other data when general users apply the tool. The impact of the following hyper-parameters on the results should be discussed in detail: a) the number of marker genes; b) the number of thresholds of the multi-Otsu thresholding algorithm; c) ω_1 and ω_2 of the MRF Prior Model; d) the scale factor of image transformation for Slide-seqV2 experiment.
5. The parameter setting for each experiment should be added. Does the experiment of Fig. 2, Fig. 4 and Fig. 5 introduce additional scRNA-seq datasets?
6. Because the Pianno method has many hyper-parameters and computational tricks, please provide a reproducible tutorial for each experiment. You can refer to STAGATE's tutorials (<https://stagate.readthedocs.io/en/latest/index.html>).
7. In Fig. 5b, how does Pianno perform background removal?
8. Pianno employ SAVER method to perform gene imputation, which makes the inputs of Pianno is different with other methods. Authors should also evaluate the clustering accuracy of other methods when using SAVER imputation results as input.
9. More deconvolution methods, such as RCTD and Tangram, should be introduced in Fig. 3. And the parameter setting of these methods should also be introduced.

Minor concerns:

1. The usage of refine/renew is confused. In the section 2.1, "refinement" is used, while "renewed" is used in Fig. 1.
2. I suggest the authors clarify the concept of "spatial domains" when it is first introduced in line 599.

Reviewer #2 (Remarks to the Author):

Zhou et al. introduced a Bayesian framework named Pianno for annotating biological structures or cell types. This framework leverages Markov random field (MRF) and parameters related to the

negative binomial distribution, estimated using the Expectation-Maximization (EM) algorithm. Pianno takes into account both gene expression levels and the physical locations of spots/cells derived from spatial transcriptomes. The process includes an initial segmentation step, where each spot/cell is assigned a label based on a predetermined set of marker genes, followed by a refinement step that utilizes the Bayesian framework to smooth out the initial segmentation.

Major comments:

1. Since Pianno employs a curated marker gene list as prior knowledge for semantic annotation on spatial transcriptomic data, the benchmarking should address two facets: (1) A comparison between marker-based supervised analysis and unsupervised analysis. (2) A contrast of semantic annotation with other extant algorithms. While the authors presented the former benchmark, the latter's direct benchmark is conspicuously absent. For an objective comparison, other methods should be given the identical gene list and data preprocessing approaches, like the SAVER denoising method employed by the authors. Moreover, considering that deep-learning methods, e.g., STAGATE, are better suited for higher-dimensional inputs, it would be better to incorporate traditional algorithms into this benchmark, such as SpatialPCA (<https://www.nature.com/articles/s41467-022-34879-1>) or the non-spatial-aware Louvain/Leiden clustering.
2. Pianno's semantic annotation should also be assessed against other gene set enrichment scoring techniques like AddModuleScore and AUCell. Incorporating a GSEA analysis could also be informative, given that SAVER was previously utilized to impute the gene expression to designed distributions.
3. The "curated marker gene list" appears central to Pianno's approach. The authors have illustrated that these markers are collated across a variety of samples (as highlighted in line 115). However, the methodology for marker collection remains nebulous. While single-cell RNA reference data can offer cell-type markers, how are anatomical structure markers collected for diverse samples? How robust is Pianno with respect to the provided gene list? The authors could consider randomly deleting or adding genes to the list for benchmarking purposes.
4. The integration of the Markov random field model and the spatial Poisson point process (sPPP) for label refinement is commendable. However, given the wide usage of the former in spatial transcriptomic analyses, a more detailed elucidation on how the sPPP model augments the Bayesian classifier's efficiency would be insightful.

Minor comments:

1. The SAVER denoising approach was initially designed for single-cell RNA data. Given that cell density is an implicit variable for the spots in spatial transcriptomic data, is the method still applicable?
2. In Fig. 1, the term "enancement" should be corrected to "enhancement".
3. Fig. 1 requires an accompanying legend for clarity.
4. From lines 193 to 195, the phrase "lack of stroma markers" is ambiguous. Does this denote an inability to locate stroma-related genes in the literature?
5. In line 432, "Bayesian classifier" has been typographically erred and requires correction.

A point-by-point response to Reviewers' comments

Enclosed is our detailed response addressing the reviewers' comments for the manuscript titled Zhou et al., NCOMMS-23-42961-T. We express our sincere appreciation to the reviewers for dedicating their time and providing insightful and constructive feedback. Their insights have significantly contributed to the enhancement of our work. We are confident that our thorough response addresses all concerns raised by the reviewers, resulting in a more impactful and improved manuscript compared to the initial submission.

In response to the comments and suggestions from both reviewers, we have made the following broad revisions:

- We enhanced the pattern detector module in Pianno by introducing "AutoPatternRecognition" and "ProposedPatterndict" functions for automatic parameter optimization and to facilitate marker selection. The revised version of Pianno now requires as few as one marker gene per pattern to start with, generating personalized updated marker list and optimized parameters for each sample, resulting in improved performance compared with the original version. All figures and text have been updated in accordance with the results obtained using the new version of Pianno.
- We assessed the performance of Pianno through benchmarking with additional tools, covering supervised approaches, unsupervised clustering approaches, and cell deconvolution methods. Furthermore, we conducted benchmarking with tools using or without SAVER for preprocessing, ensuring fair and comprehensive comparisons.
- We evaluated the sensitivity of Pianno to the choice of marker genes.
- We provided reproducible tutorials, along with the required configuration files and tutorial documentation for reference, accessible at GitHub.

A complete description of the changes we have made in response to reviewer comments is detailed below.

Reviewer #1

In this manuscript, the authors developed a new computational model, Pianno, to identify structural semantics based on marker genes. The authors showed that Pianno can estimate cell type distributions and annotate a wide array of spatial semantics. Actually, the pattern image generated by the pattern detector module of Pianno is a linear combination of the denoised expression of marker genes, which makes the selection of marker genes particularly important. However, how to select the marker gene list and parameter settings are not described in the paper, and its robustness and scalability need to be further validated.

Response: We sincerely appreciate dedication and meticulous effort in evaluating our manuscript. We believe that the insightful comments and suggestions provided are invaluable in elevating the overall quality of our work. In response to the raised suggestions, we have diligently implemented the following improvements and would like to share our detailed responses.

Major concerns:

1. The selection process for marker genes should be described in detail and avoid manual selection. The core step of Pianno is to obtain the pattern image by linearly fusing the expression of marker genes. But how to select the marker gene list is not described in the paper. Authors should ensure that this step is fully automated.

Response: We appreciate the reviewer's thoughtful consideration and constructive suggestions. Pianno, designed to replicate the manual labeling process for known structures on spatial transcriptomes, relies on some prior information, such as classical markers from immunostaining, to match with these known structures. We acknowledge the importance of offering a detailed description of the marker gene selection process and implement an automatic approach to facilitate the process, thereby enhancing Pianno's generability, usability, and overall user-friendliness.

In response to this valuable suggestion, we have enhanced the pattern detector module in Pianno. The updated module now requires minimal prior information, with users specifying as few as one known marker for each pattern, and allowing up to one pattern to have no initial marker. The improved model generates a list of additional candidate markers, and automatically optimizes hyperparameter selection.

Specifically, we introduced "AutoPatternRecognition" and "ProposedPatterndict" functions for automatic parameter optimization and to facilitate marker selection. First, "AutoPatternRecognition" optimizes the combination of parameters for initial annotation based on a single known marker for each pattern provided by users. Next, the "ProposedPatterndict" function outputs a selected number of candidate marker genes for each pattern, ranked by their enrichment in the designed pattern according to the initial annotation generated by "AutoPatternRecognition". The number of candidate markers can be adjusted by users, and fewer genes will be output if the number of marker genes identified is fewer than the specified number. From the candidate marker list, users can select 1-3 markers based on their expression patterns to form the updated marker list for further usage. The updated marker list and the optimized parameters are then used to update initial pattern detection. Subsequently, the updated marker list and the updated initial annotation are fed into the Bayesian classifier to calculate posterior probabilities (Methods 4.2.5).

Prior to these improvements, we used the same marker list and parameters for pattern detection across samples, which posed robustness challenges. The incorporation of the automatic optimization process now allows personalized marker list and parameter selection for each sample, resulting in improved annotation performance (Fig. 2b).

2. Is Pianno sensitive to the choice of marker gene list? Will performance degrade if 10% or 20% of the genes in the marker gene list are randomly selected?

Response: To assess Pianno's sensitivity to the choice of marker gene lists, we generated a total marker gene list containing 10 markers for each pattern. This list was constructed based on the union sets of the top 10 candidate marker genes selected from each sample by the newly added "ProposedPatterndict" function, starting with a fixed initial marker list with one marker per pattern (refer to Methods for details). Subsequently, we evaluated Pianno's performance by varying the number of marker genes per pattern randomly selected from the total list. Notably, we observed an improvement in Pianno's accuracy with an increasing number of markers, surpassing a commendable accuracy (> 0.6) when the number of markers per pattern exceeded 6 (Fig. S6). This

trend is likely attributed to the inclusion of noise marker genes in each sample, where certain genes may rank among the top marker genes in one sample but fail to exhibit pattern-specific expression in other samples due to technical variations like dropouts.

To further explore the impact of noise inclusion in the marker list on Pianno's accuracy, we calculated the percentage of noise genes—those not included in the top 10 marker list of a given sample when using the total list. The average percentage of noise genes was found to be 47%, and yet the average accuracy still achieved 0.71. This suggests that the inclusion of noise genes may be compensated for by the increased number of genes, presenting a higher chance to incorporate high-quality markers. In contrast, selecting a small number (1-3) of personalized markers for each sample resulted in an enhanced accuracy of Pianno (Fig. S6 & Methods 4.5).

In summary, our results underscore the importance of the inclusion of high-quality markers, which can be achieved by either increase the number of top marker genes with potential noise, or by adopting individualized marker selection for each sample starting with minimal prior information, as initially suggested by the reviewer in Question #1.

3. Pianno stores gene expression in the form of images (i.e., dense matrix). Can Pianno handle large-scale data? If scale factor is introduced, is it equivalent to reducing the spatial resolution of the sequencing technology?

Response: We appreciate the reviewer's thoughtful inquiry. Pianno is built upon the prevailing data storage structure, Anndata, which stores gene expression data in the form of a sparse matrix. In this structure, only a pattern image and a feature image are stored as dense matrices. Based on current experimental results, Pianno is capable of handling large-scale data such as Stereo-seq and Slide-seq.

We acknowledge that the introduction of the scale factor does reduce the resolution of the technology. For Stereo-seq, where the low gene numbers are captured on each nano-particle, we employed a scale factor to bin spots, using Bin 50 with a 25um resolution, compromising the sequencing depth. This was necessary to accommodate the unique characteristics of Stereo-seq.

Similarly, for Slide-seq, the microparticles capturing RNAs are sparse and not organized in rows and columns as in Stereo-seq or 10X spatial transcriptomics. It also has low sequencing depth like in Stereo-seq. Therefore, we binned the spots to adapt to the sparse and irregular organization of microparticles in Slide-seq.

4. The algorithm may potentially be affected by “p-hacking”, where parameters/settings are tuned to get the best performance of the reported data. As the study did not use many datasets, such a good performance may not be transferable to other data when general users apply the tool. The impact of the following hyper-parameters on the results should be discussed in detail: a) the number of marker genes; b) the number of thresholds of the multi-Otsu thresholding algorithm; c) ω_1 and ω_2 of the MRF Prior Model; d) the scale factor of image transformation for Slide-seqV2 experiment.

Response: We appreciate the reviewer's constructive suggestions. We designed three control variable experiments using the dIPFC dataset to investigate the impacts of (a) the number of marker genes; (b) the number of thresholds of the multi-Otsu thresholding algorithm; and (c) ω_1 and ω_2 of

the MRF Prior Model in Methods 4.5. (d) We have also demonstrated the effects of size factor selection below, and added a practical guidance for size factor selection in the tutorial.

Specifically:

- (a) We first constructed a total marker list of 10 markers per pattern (Supplementary Material) that appear in the top 10 candidate marker genes for at least one dIPFC sample. For each trial, we fixed the other parameters, while randomly selecting N (N increments from 1 to 10) genes from the total marker list. The accuracy of Pianno annotation was calculated and repeated three times for each sample. It seems that as the number of markers increased, the accuracy exhibited a positive correlation. Notably, manually selecting 1-3 marker genes from the top 10 marker gene list, guided by careful inspection of gene patterns, further elevated Pianno's performance (Supplementary Fig. S6a). This experiment shows the quantity of markers is not as many as the better, and quality is more important. We suggest selecting 1-3 robustly expressed genes for each pattern when constructing the marker list.
- (b) While maintaining other parameters unchanged and utilizing a manually curated marker list, the performance of Pianno was evaluated by systematically increasing the values of thresholds in the Multi-Otsu thresholding algorithm from 1 to 5. The accuracy of Pianno's annotations for each sample was then calculated. Notably, when n equaled 1, no thresholding segmentation is performed, and for excessively large n values ($n > 5$), segmentation was impractical due to restricted gene expression value ranges. The optimal performance of Pianno was observed when n equaled 2 or 3, reflecting the states of presence and absence or categorization into high, medium, and low gene expression levels, respectively (Supplementary Fig. S6b).
- (c) While keeping other parameters constant, the performance of Pianno was assessed by increasing the values of ω_1 and ω_2 from 0 to 1 with a step size of 0.01. This was done to calculate the accuracy of Pianno's annotation specifically for sample 151673. The hyperparameters ω_1 represents the effects of transcriptome and spatial similarity, while ω_2 represents the global consistency on the prior distribution. The experimental results indicated that a larger ω_1 combined with a smaller ω_2 achieved the best annotation results (Supplementary Fig. S6c). It's noteworthy that, in studies in this manuscript, we default to using $\omega_1 = 0.99$ and $\omega_2 = 0.01$.
- (d) In the manuscript, we use size factors of 1, 50, and 30 for 10X spatial transcriptomics, Stereo-seq, and Slide-seq V2, respectively, to balance resolution and sequencing depth. The figure below provides an example of mask images obtained under various size factors on a Slide-seq V2 hippocampus sample. For practical usage, we offer guidance on size factor selection in the tutorial.

5. The parameter setting for each experiment should be added. Does the experiment of Fig. 2, Fig. 4 and Fig. 5 introduce additional scRNA-seq datasets?

Response: The experiments depicted in Fig. 2, Fig. 4, and Fig. 5 did not involve the incorporation of additional scRNA-seq datasets. All marker genes utilized for Pianno were sourced from published literature. To ensure transparency and reproducibility, we have uploaded reproducible tutorials and configuration files to GitHub (<https://github.com/yuqiuzhou/Pianno>). These materials provide detailed information on parameter settings and marker gene lists for all experiments featured in the manuscript.

6. Because the Pianno method has many hyper-parameters and computational tricks, please provide a reproducible tutorial for each experiment. You can refer to STAGATE's tutorials (<https://stagate.readthedocs.io/en/latest/index.html>).

Response: We sincerely value the reviewer's constructive suggestion. In order to facilitate reproducibility and usage, we have provided reproducible tutorials for each experiment. These tutorials, along with the required configuration files, are available on GitHub (<https://github.com/yuqiuzhou/Pianno>). Additionally, we have created tutorial documentation for reference, accessible at (<https://pianno-tutorials.readthedocs.io/en/latest/index.html>). This documentation offers detailed insights into the implementation of Pianno, akin to the tutorials provided by STAGATE. We hope these resources will contribute to a clear understanding and reproducibility of our methodology, and promote its usage.

7. In Fig. 5b, how does Pianno perform background removal?

Response: Pianno labels spots without assignments to any specified category with given markers as "undefined". Users can subsequently categorize this "undefined" label either as background or as a novel, previously unknown category. In Fig. 5b, there is no background removal involved.

8. Pianno employ SAVER method to perform gene imputation, which makes the inputs of Pianno is different with other methods. Authors should also evaluate the clustering accuracy of other methods when using SAVER imputation results as input.

Response: We appreciate the reviewer's insight into the potential differences in inputs compared to other methods. To facilitate objective comparisons, we evaluated the Adjusted Rand Index (ARI)

changes for other methods with or without gene imputation by SAVER, except for CellAssign, which only accepts raw counts as input. The results, as depicted in Fig. 2c, indicate that, except for SCANPY, employing denoised inputs with other methods does not enhance clustering accuracy. In some cases, it even yields results significantly inferior to using raw counts directly (as observed in STAGATE and SpaGCN).

9. More deconvolution methods, such as RCTD and Tangram, should be introduced in Fig. 3. And the parameter setting of these methods should also be introduced.

Response: In the revised version, we compared the results from Pianno with more deconvolution methods, including Cell2Location, Tangram, and RCTD in Fig. S3b. Pianno demonstrated comparable performance to Tangram and RCTD, and outperformed Cell2Location. Additionally, we have provided detailed information about the parameter settings of these methods in the Method section.

Minor concerns:

10. The usage of refine/renew is confused. In the section 2.1, “refinement” is used, while “renewed” is used in Fig. 1.

Response: In section 2.1, we aim to convey that the label "renewed" is derived from the "refinement" step. Recognizing the potential confusion in this expression, we have opted to substitute the term "Renewed annotation" with "Final annotation" in Fig. 1.

11. I suggest the authors clarify the concept of “spatial domains” when it is first introduced in line 599.

Response: We have provided an additional clarification for the term in the revised manuscript, elaborating on it in Methods 4.6.

Reviewer #2

Zhou et al. introduced a Bayesian framework named Pianno for annotating biological structures or cell types. This framework leverages Markov random field (MRF) and parameters related to the negative binomial distribution, estimated using the Expectation-Maximization (EM) algorithm. Pianno takes into account both gene expression levels and the physical locations of spots/cells derived from spatial transcriptomes. The process includes an initial segmentation step, where each spot/cell is assigned a label based on a predetermined set of marker genes, followed by a refinement step that utilizes the Bayesian framework to smooth out the initial segmentation.

Response: We appreciate the reviewer's comprehensive and constructive suggestions. In response to the questions raised by the reviewer, we offer the following responses.

Major concerns:

1. Since Pianno employs a curated marker gene list as prior knowledge for semantic annotation on spatial transcriptomic data, the benchmarking should address two facets: (1) A comparison between marker-based supervised analysis and unsupervised analysis. (2) A contrast of semantic annotation with other extant algorithms. While the authors presented the former benchmark, the latter's direct

benchmark is conspicuously absent. For an objective comparison, other methods should be given the identical gene list and data preprocessing approaches, like the SAVER denoising method employed by the authors. Moreover, considering that deep-learning methods, e.g., STAGATE, are better suited for higher-dimensional inputs, it would be better to incorporate traditional algorithms into this benchmark, such as SpatialPCA (<https://www.nature.com/articles/s41467-022-34879-1>) or the non-spatial-aware Louvain/Leiden clustering.

Response: We appreciate the constructive suggestion provided by the reviewer regarding benchmarking. In response, we extended our comparisons by incorporating algorithms belonging to different categories:

(1) Comparison with marker-based supervised methods: Given that Pianno is the first marker-based semantic annotation method for spatial data, we introduced CellAssign, another marker-based cell-type annotation method designed for single cells, into the benchmark. Both methods were evaluated using an identical marker gene list. The observations revealed that the non-spatially aware CellAssign struggled to semantically annotate spatial transcriptomes effectively. Additionally, it did not perform as well as Pianno, particularly when dealing with a small number of markers (1-3 markers per pattern) (Fig. 2).

(2) Comparison with classical non-deep-learning methods: We included the most widely used non-spatially aware traditional unsupervised approach—the Leiden clustering algorithm implemented in SCANPY. Pianno exhibited superior performance over this traditional unsupervised clustering approach (Fig. 2).

(3) Comparisons across tools with noise reduction preprocessing: To facilitate fair comparisons between methods using the same preprocessing steps, we calculated the Adjusted Rand Index (ARI) changes for other tools with or without noise reduction. Notably, CellAssign was excluded from this analysis as it solely accepts raw counts as input. The results, as illustrated in Fig. 2c, indicate that, except for SCANPY, which exhibited improved performance but still scored lower than Pianno with preprocessing, other methods using denoised inputs failed to improve the performance. In certain cases, their performance even deteriorated compared to using raw counts, as evidenced in the case of STAGATE and SpaGCN (Fig. 2c).

2. Pianno's semantic annotation should also be assessed against other gene set enrichment scoring techniques like AddModuleScore and AUCell. Incorporating a GSEA analysis could also be informative, given that SAVER was previously utilized to impute the gene expression to designed distributions.

Response: We appreciate the reviewer's thoughtful consideration and constructive suggestions. For dIPFC Sample 151507, we have computed the AddModuleScore and AUCell scores using the same marker gene list generated by the newly implemented automatic marker gene selection module in the revised version of Pianno (please refer to Question #3). While AddModuleScore exhibits good specificity, it appears sparse for certain patterns. In contrast, AUCell demonstrates overlap between patterns, as depicted in the figure below. Therefore, we have decided not to incorporate these scoring techniques into the Pianno pipeline.

Regarding the suggestion for a Gene Set Enrichment Analysis (GSEA), which typically involves pathway enrichment analysis, we are currently unsure about the optimal approach for integrating it

into our methodology. We would greatly appreciate further guidance on how to effectively incorporate GSEA analysis into Pianno.

3. The “curated marker gene list” appears central to Pianno's approach. The authors have illustrated that these markers are collated across a variety of samples (as highlighted in line 115). However, the methodology for marker collection remains nebulous. While single-cell RNA reference data can offer cell-type markers, how are anatomical structure markers collected for diverse samples? How robust is Pianno with respect to the provided gene list? The authors could consider randomly deleting or adding genes to the list for benchmarking purposes.

Response: We appreciate the constructive suggestions provided by the reviewer. Recognizing the pivotal role of marker gene selection in Pianno's approach, we have made significant enhancements of the Pianno tool and provided corresponding changes in the current version of our manuscript. In the previous version, we manually selected classical markers from literature, other region-specific bulk RNA-seq data, or immunostaining studies for known structure mappings. The revised version now includes a detailed description of the marker gene selection process, along with the implementation of an automatic approach aimed at streamlining the marker selection process, thereby improving Pianno's generability, usability, and overall user-friendliness.

To achieve this, we have revamped the pattern detector module in Pianno. The updated module now requires minimal prior information, allowing users to specify as few as one known marker for each pattern, with the flexibility of permitting up to one pattern to have no initial marker. We have enhanced the pattern detector module in Pianno. The updated module now requires minimal prior information, with users specifying as few as one known marker for each pattern, and allowing up to one pattern to have no initial marker. The improved model generates a list of additional candidate markers, and automatically optimizes parameter selection.

Two key functions, "AutoPatternRecognition" and "ProposedPatternDict," have been introduced to facilitate automatic parameter optimization and marker selection. In the "AutoPatternRecognition" function, the combination of parameters for initial annotation is optimized based on a single known marker provided by users for each pattern. Subsequently, the "ProposedPatternDict" function outputs a selected number of candidate marker genes for each pattern, ranked by their enrichment in the

designed pattern according to the initial annotation generated by "AutoPatternRecognition." Users have the flexibility to adjust the number of candidate markers, and fewer genes will be output if the identified marker genes fall below the specified number. From this list, users can then select 1-3 markers based on their expression patterns to form an updated marker list for further usage. The updated marker list and optimized parameters are then employed to update the initial pattern detection. Following this, the updated marker list and initial annotation are fed into the Bayesian classifier to calculate posterior probabilities (Methods 4.2.5).

Prior to these improvements, we used the same marker list and parameters for pattern detection across samples, which posed robustness challenges. The incorporation of the automatic optimization process now allows personalized marker list and parameter selection for each sample, resulting in improved annotation performance (Fig. 2b).

In evaluating Pianno's sensitivity to the choice of marker gene lists, we generated a total marker gene list containing 10 markers for each pattern. This list was constructed based on the union sets of the top 10 candidate marker genes selected from each sample using the newly added "ProposedPatternDict" function, starting with a fixed initial marker list with one marker per pattern (Methods 4.5). Subsequently, we assessed Pianno's performance by varying the number of marker genes per pattern randomly selected from the total list. Notably, an improvement in Pianno's accuracy was observed with an increasing number of markers, surpassing a commendable accuracy (> 0.6) when the number of markers per pattern exceeded 6, as shown in Fig. S6. This trend is attributed to the inclusion of noise marker genes in each sample, where certain genes may rank among the top marker genes in one sample but fail to exhibit pattern-specific expression in other samples due to technical variations like dropouts. Additionally, selecting a small number (1-3) of high-quality personalized markers for each sample from the top 10 marker gene list resulted in further enhanced accuracy for Pianno, as illustrated in Fig. S6.

Overall, the assessment underscores the importance of the inclusion of high-quality markers, achievable by either increase the number of top marker genes with potential noise, or by adopting personalized marker selection for each sample starting with minimal prior information, as facilitated by our new model.

4. The integration of the Markov random field model and the spatial Poisson point process (sPPP) for label refinement is commendable. However, given the wide usage of the former in spatial transcriptomic analyses, a more detailed elucidation on how the sPPP model augments the Bayesian classifier's efficiency would be insightful.

Response: We thank the reviewer think the integration of the MRF and sPPP for label refinement is commendable. In response to the inquiry about how the sPPP model enhances the efficiency of the Bayesian classifier, we selected sPPP for its ability to model count data from RNA-seq, considering the covariance between spatially neighboring spots. We believe that the augmentation in our model comes from the following key aspects: 1) the consideration of marker genes as a pseudo-image provides a robust prior distribution for the MRF, contributing to the label refinement process; 2) In the subsequent MRF design, we take into account both transcriptomic and spatial similarities, along with the global consistency between spots. This comprehensive approach ensures an accurate refinement of labels.

To provide further clarity, we have included a paragraph in the discussion section to elaborate on these aspects and their contributions to the enhanced efficiency of our model.

Minor concerns:

5. *The SAVER denoising approach was initially designed for single-cell RNA data. Given that cell density is an implicit variable for the spots in spatial transcriptomic data, is the method still applicable?*

Response: While the SAVER denoising approach was initially tailored for single-cell RNA data, we have observed its applicability and effectiveness in the context of denoising spatial transcriptomics in our model. Our method benefits from SAVER's unique ability to leverage gene-to-gene relationships for recovering the true expression level of each gene in every spot. Unlike alternative approaches that rely on similarities across cells, SAVER's design suggests that the implicit variable of cell density in each spot does not inherently impact its performance. Hence, we choose SAVER for denoising spatial transcriptome data in our model

6. *In Fig. 1, the term “enancement” should be corrected to “enhancement”.*

Response: The mentioned typo has been corrected.

7. *Fig. 1 requires an accompanying legend for clarity.*

Response: We have included a legend for *Fig. 1* in the revised version.

8. *From lines 193 to 195, the phrase “lack of stroma markers” is ambiguous. Does this denote an inability to locate stroma-related genes in the literature?*

Response: The identification of stroma markers is challenging given the complex composition of stromal tissue. Stromal elements encompass diverse cell types such as fibroblasts and endothelial cells, each associated with distinct markers. Considering this ambiguity, we refrained from specifying stroma markers in our analysis. Instead, we designated spots assigned to the "undefined" category as stromal regions. We have revised the original sentence to reflect this clarification.

9. *In line 432, “Bayesian classifier” has been typographically erred and requires correction.*

Response: The correction has been made.

Reviewer #1 (Remarks to the Author):

Thanks for the response. Most of my concerns have been addressed. The updated pattern detector module of Pianno can use only one marker as prior knowledge, and provides detailed tutorials, which greatly improves the usability of the algorithm.

Reviewer #1 (Remarks on code availability):

I installed and run the code and authors provided detailed tutorials.

Reviewer #2 (Remarks to the Author):

After the authors detailed the process for selecting markers to identify spatial domains, the methodology now appears complete and clear, and I have no further comments on the results section. I acknowledge its high effectiveness across multiple tissues as demonstrated by the authors. However, concerns arise regarding its generalizability to tissues with less-studied molecular structures. The software's performance heavily depends on the initial selection of markers, which need to be highly aligned with and specifically expressed in the corresponding domain. This approach restricted the analysis primarily to cell types in the two cancer samples demonstrated, rather than to the tumor niches. Consequently, this implies that the molecular signature must be reliably documented for the anatomical structure in the initial step, limiting the algorithm's potential to uncover unknown biological patterns.

A point-by-point response to Reviewers' comments

Reviewer #1

Thanks for the response. Most of my concerns have been addressed. The updated pattern detector module of Pianno can use only one marker as prior knowledge, and provides detailed tutorials, which greatly improves the usability of the algorithm.

Response: The reviewer's positive feedback is greatly appreciated, and we are committed to continually improving the algorithm for better usability and effectiveness.

Reviewer #2

After the authors detailed the process for selecting markers to identify spatial domains, the methodology now appears complete and clear, and I have no further comments on the results section. I acknowledge its high effectiveness across multiple tissues as demonstrated by the authors. However, concerns arise regarding its generalizability to tissues with less-studied molecular structures. The software's performance heavily depends on the initial selection of markers, which need to be highly aligned with and specifically expressed in the corresponding domain. This approach restricted the analysis primarily to cell types in the two cancer samples demonstrated, rather than to the tumor niches. Consequently, this implies that the molecular signature must be reliably documented for the anatomical structure in the initial step, limiting the algorithm's potential to uncover unknown biological patterns.

Response: We appreciate the reviewer's insights and concerns regarding the generalizability of Pianno to tissues with less-studied molecular structures. We added a sentence in Discussion to declare the potential limitations of Pianno. Here is the updated statement:

“While Pianno has demonstrated remarkable power in spatial semantic annotation, it is imperative to acknowledge that its efficacy is inherently linked to the availability of well-defined initial markers and the existing molecular knowledge of the tissue, which may limit the algorithm's ability to uncover unknown biological patterns.”